# Extreme environmental conditions reduce coral reef fish biodiversity and productivity

Simon J. Brandl [1,2,3,4,10✉], Jacob L. Johansen [5,6,10✉], Jordan M. Casey[3,4], Luke Tornabene[7], Renato A. Morais [8,9] & John A. Burt [6]

Tropical ectotherms are hypothesized to be vulnerable to environmental changes, but cascading effects of organismal tolerances on the assembly and functioning of reef fish communities are largely unknown. Here, we examine differences in organismal traits, assemblage structure, and productivity of cryptobenthic reef fishes between the world's hottest, most extreme coral reefs in the southern Arabian Gulf and the nearby, but more environmentally benign, Gulf of Oman. We show that assemblages in the Arabian Gulf are half as diverse and less than 25% as abundant as in the Gulf of Oman, despite comparable benthic composition and live coral cover. This pattern appears to be driven by energetic deficiencies caused by responses to environmental extremes and distinct prey resource availability rather than absolute thermal tolerances. As a consequence, production, transfer, and replenishment of biomass through cryptobenthic fish assemblages is greatly reduced on Earth's hottest coral reefs. Extreme environmental conditions, as predicted for the end of the 21st century, could thus disrupt the community structure and productivity of a critical functional group, independent of live coral loss.

[1] Department of Biological Sciences, Simon Fraser University, Burnaby, BC, Canada. [2] CESAB—FRB, 5 Rue de l'École de Médecine, 34000 Montpellier, France. [3] PSL Université Paris: CNRS-EPHE-UPVD USR3278 CRIOBE, Université de Perpignan, Perpignan, France. [4] Laboratoire d'Excellence "CORAIL,", Perpignan, France. [5] Hawai'i Institute of Marine Biology, University of Hawai'i at Manoa, Kane'ohe, HI, USA. [6] Marine Biology Laboratory, Centre for Genomics and Systems Biology, New York University Abu Dhabi, Abu Dhabi, United Arab Emirates. [7] School of Aquatic and Fishery Sciences and the Burke Museum of Natural History and Culture, University of Washington, Seattle, WA, USA. [8] ARC Centre of Excellence for Coral Reef Studies, James Cook University, Townsville, QLD, Australia. [9] College of Science and Engineering, James Cook University, Townsville, QLD, Australia. [10]These authors contributed equally: Simon J. Brandl, Jacob L. Johansen. ✉email: simonjbrandl@gmail.com; jacob.johansen@hawaii.edu

Why do some species occur in a given location while similar taxa are missing? And how do resulting assemblages affect rates of ecological processes? As escalating human impacts on the biosphere deplete and re-shuffle biological communities across ecosystems[1,2], answers to these questions are key to our quest to preserve biodiversity and eco-system services to humanity[3].

A species' presence at a given location is mediated by a hierarchical interplay between organismal traits (e.g., temperature tolerance, trophic niche), environmental conditions (e.g., temperature, salinity, dissolved oxygen), biotic interactions (e.g. habitat or food availability), biogeographic history, and stochastic events (e.g., extinction, dispersal)[4–6]. Furthermore, the identity and diversity of species impact rates of ecosystem functioning, including processes that are critical to human well-being, such as primary or consumer productivity[7,8]. However, by modifying abiotic conditions, species' niches, and biotic interactions, global stressors such as climate change can interfere with these dynamics through numerous pathways[9–11]. At the organismal level, changes in environmental factors, such as temperature, affect internal physiological processes in ectotherms (e.g., oxygen consumption)[12], which, if not lethal, will alter organismal energy expenditure[13–15]. Changes in organismal energy demands subsequently drive resource acquisition (e.g. feeding rates, prey species) and how resulting energy is allocated to life-supporting processes (homeostasis), growth, and reproduction[16–18]. Dynamics of energy acquisition and investment, which are often investigated through the lens of ecological niches and fitness, are the basis of modern coexistence theory and critical for our understanding of community assembly dynamics[19] and the rate of ecological processes that underpin energy and nutrient fluxes through ecosystems[20]. Integration across levels of biological organization is, therefore, crucial to understand the effects of global environmental change on our planet's ecosystems[21].

Coral reefs are the most diverse marine ecosystem, and their productivity provides vital services for more than 500 million people worldwide[22]. Scleractinian corals, the foundation species of tropical reefs, show high sensitivity to thermal extremes, which has led to the rapid global decline of coral reef ecosystems[23]. In the wake of losing coral habitat, communities of the most prominent reef consumers, teleost fishes, also decline or shift in composition[24–27], which directly affects the provision of resources to people dependent on reef fisheries[28]. Although recent evidence suggests that some fish species will be able to cope with (or even benefit from) live coral loss, at least in the short term[28–31], tropical reef fishes are typically adapted to a relatively narrow suite of environmental conditions. Thus, reef fishes may also be vulnerable to the direct effects of climate change on, for instance, sea surface temperatures[13,32,33]. Consequently, the responses of reef fishes to ongoing changes in their environment might be as important as indirect, habitat-mediated responses[34–36].

While other environmental factors (such as salinity or oxygen saturation) have considerable effects on reef fish physiology[37], temperature is by far the most commonly investigated environmental stressor for reef fishes. Despite marked differences in species-specific tolerances to higher temperatures[38–43], most reef fish species suffer from non-lethal[44] adverse physiological, developmental, or behavioral responses when exposed to temperatures outside their normal range. Current understanding suggests long-term deleterious effects on reef fish populations in the wild[34], but few cases of direct temperature-mediated population declines have been documented in situ for reef fish communities[45]. One factor that ameliorates the adverse effects of rising temperatures in the wild may be transgenerational plasticity, which can enhance the performance of offspring in higher temperatures through developmental, genetic, or epigenetic pathways[36,46]. This has been shown in a few model species[36,46,47], but demands increased energetic investments[46,48]. It is unclear whether this process can truly enhance the survival of reef fishes in competitive, uncontrolled environments, and how species-specific temperature tolerance differences may mediate coexistence in ecological communities.

Cryptobenthic fishes are the smallest of all reef fishes, rarely exceeding 50 mm in maximum body size[49]. They account for almost half of all reef fish species and are numerically abundant and ubiquitous on reefs worldwide[49–52]. Due to their small body size, these fishes have evolved a unique life-history strategy of rapid growth, high mortality, and continuous larval replenishment, and play an important role in coral reef trophodynamics[53]. Their small body size and associated life-history also promise exceptional traceability concerning the effects of, and responses to, changing environmental conditions[49]. Limited gill surface area, high mass-specific metabolism, and other physiological challenges resulting from their minute size suggest that cryptobenthics are particularly susceptible to temperature fluctuations[40,49,54]. Due to their limited mobility and close association with the benthos[55], mitigation of temperature extremes through migration is also not viable and notable shifts in cryptobenthic fish community composition have been observed following small-scale changes in the benthic community structure[27,56]. However, the extremely high generational turnover (7.4 generations per year in the most extreme species[53,57]) and prevalence of benthic clutch spawning and parental care[49] may make them ideally suited for transgenerational adaptation to changing conditions[34]. In fact, an extremely fast evolutionary clock has been implicated as a driver for rapid speciation in cryptobenthic fishes[58], which may permit similarly fast microevolutionary changes (i.e., rapid adaptation). Thus, cryptobenthic fishes may be well-suited to detect the impact of environmental change on organisms and populations, with promising insights into whether transgenerational plasticity or adaptation can provide pathways to the persistence of coral reef fishes in changing oceans.

Here, we quantify cryptobenthic community structure, species- and population-specific physiological and dietary traits, and contributions to ecosystem functioning in the world's hottest, most extreme coral reef environment, the southeastern Arabian Gulf, and we compare the resulting patterns with the spatially proximate, but more thermally moderate, Gulf of Oman. Specifically, the goal of our study was to (1) describe cryptobenthic fish assemblages across the two locations, (2) identify organismal traits that permit or preclude existence in the Arabian Gulf, and (3) determine the consequences of these results for the production, provision, and renewal of cryptobenthic fish biomass[21]. We show that cryptobenthic fish assemblages in the Arabian Gulf are much less diverse and abundant compared to the nearby Gulf of Oman. Yet, thermal tolerances of cryptobenthic fish species indicate that all species found in the Gulf of Oman should be able to withstand the thermal extremes of the southern Arabian Gulf. However, population-specific differences in ingested prey and body condition in three species that occur in both locations suggest that the environmental extremes of the southern Arabian Gulf foster an energetically challenging environment that precludes the presence of many cryptobenthic species. As a consequence, the production, transfer, and renewal of biomass through cryptobenthic fish communities are severely compromised on reefs in the southern Arabian Gulf.

## Results

**Environmental context**. Between 2010 and 2018, remotely sensed temperature data[59] from the studied sites ranged between

a minimum of 19.1 °C (Gulf of Oman in 2016) and a maximum of 32.9 °C (Arabian Gulf in 2014) (Supplementary Fig. 1), with the seven highest temperatures all occurring in the Arabian Gulf. In situ data loggers deployed on the substrate (4–6 m depth) at our study sites[60] recorded summer maximum temperatures of 36.0 °C (mean daily maximum from 2012 to 2017: 33.7 °C) in the Arabian Gulf and 34.8 °C (mean daily maximum from 2012 to 2014: 29.9 °C) in the Gulf of Oman, while recording minimum winter temperatures of 17.3 °C (mean daily minimum = 22.0 °C) in the Arabian Gulf and 21.5 °C (mean daily minimum = 23.7 °C) in the Gulf of Oman (Supplementary Fig. 2). Moreover, in 2012, 2013, and 2014 (for which data were available from both locations), sites in the Arabian Gulf recorded an average of 69.0, 63.7, and 64.3 days per year, respectively, where daily maximum temperatures exceeded 34 °C, while sites in the Gulf of Oman recorded averages of 0.0, 1.0, and 5.0 days, respectively (Supplementary Table 1)[61]. Thus, maximum temperatures on reefs along the Arabian Gulf coast of the United Arab Emirates closely approach forecasted temperatures for tropical coral reefs at the end of the century[62]. While the two locations also differ in several co-varying environmental factors, including salinity, productivity, or reef geomorphology, temperature is commonly considered the strongest environmental force that shapes life in the Arabian Gulf[60–64]. Despite the seemingly unfavorable conditions for tropical reef building corals in the past and present, corals have persisted in this region for approximately 15,000 years, with the modern coastline harboring coral reef structures for circa 6000 years[62]. Therefore, the Arabian Gulf represents a useful natural laboratory to examine the capacity of reef organisms to cope with extreme environmental conditions (particularly high temperatures) and how this influences the diversity and ecological dynamics that underpin modern coral reefs (Fig. 1a, b).

**Cryptobenthic fish assemblages.** Cryptobenthic reef fish assemblages markedly differed between the Arabian Gulf and the Gulf of Oman. Reefs in the Arabian Gulf, on average, harbored less than half the richness ($1.62 \pm 0.01$ SE species m$^{-2}$ vs. $3.40 \pm 0.26$ SE species m$^{-2}$; Bayesian hierarchical model estimate: *Gulf of Oman*: $\beta = 0.73$ [0.44, 1.01; lower and upper 95% credible interval]) and less than a quarter of the abundance ($6.12 \pm 1.09$ SE individuals m$^{-2}$ vs. $31.94 \pm 1.49$ SE individuals m$^{-2}$; *Gulf of Oman*: $\beta = 1.77$ [1.03, 2.58]) of cryptobenthic fishes (Fig. 1c, d), but standing biomass estimates were comparable (*Gulf of Oman*: $\beta = 0.63$ [$-0.54$, 1.71]; Fig. 1e). Similarly, the composition of cryptobenthic communities greatly varied between the two locations (Fig. 2a), with no overlap among convex hull polygons in the nonmetric multidimensional scaling (nMDS) ordination and a strong effect of *Location* in the PERMANOVA using a site-by-species dissimilarity matrix (*Location*: d.f. = 1, $F = 13.58$, $P = 0.001$, $R^2 = 0.46$). There were 13 unique species in the Arabian Gulf, 29 unique species in the Gulf of Oman, and 16 species shared among the two locations. Importantly, of the 29 unique Gulf of Oman species, 89.7% have been recorded from the northern Arabian Gulf in Kuwait and Saudi Arabia (but not the southeastern region), where summer conditions are much less extreme[65] (Fig. 1; Supplementary Table 2). In contrast to the cryptobenthic fish community, there were no statistical differences in coral cover (Bayesian hierarchical model: *Gulf of Oman*: $\beta = 0.02$ [$-1.30$, 1.42]) nor overall benthic community structure as revealed by a PERMANOVA (*Location*: d.f. = 1, $F = 1.63$, $P = 0.187$, $R^2 = 0.09$; Fig. 2b). Thus, despite broadly comparable benthic community composition and live coral cover (two commonly quantified metrics), cryptobenthic fish assemblages strongly differed between the two locations.

**Temperature tolerance.** We tested whether organismal temperature tolerance can explain the absence of three abundant Gulf of Oman species (*Helcogramma fuscopinna*, *Eviota guttata*, and *Hetereleotris vulgaris*) from the thermally extreme southeastern Arabian Gulf, despite their recorded presence in more moderate parts of the Arabian Gulf. We also examined the potential for intraspecific differences in thermal tolerance in two species with populations in both locations (*Enneapterygius ventermaculus* and *Ecsenius pulcher*) and examined the Arabian Gulf population of an additional species for which we were unable to obtain enough samples from the Gulf of Oman (*Coryogalops anomolus*). Species-specific critical thermal tolerance limits did not explain the absence of three common Gulf of Oman species in the Arabian Gulf (Fig. 3). The mean critical thermal maximum tolerance limits ($CT_{max}$) of all six tested species, regardless of origin, equaled or surpassed the maximum summer temperatures recorded in the Arabian Gulf (36.0 °C). *Helcogramma fuscopinna* (a Gulf of Oman species) had the lowest heat tolerance at $36.0 \pm 0.11$ °C, while *C. anomolus* from the Arabian Gulf had the greatest heat tolerance ($38.4 \pm 0.06$ °C). While there were no population differences in heat tolerance for *E. ventermaculus* (possibly due to limited samples from the Gulf of Oman), the Arabian Gulf population of *E. pulcher* showed slightly greater heat tolerance (0.6 °C) than their Gulf of Oman counterparts ($37.9 \pm 0.05$ °C SE vs. $37.3 \pm 0.06$ °C SE), providing some evidence for enhanced thermal tolerance in this species. Despite considerable interspecific differences and some evidence for intraspecific thermal plasticity (Supplementary Table 3), mean predicted maximum posterior heat tolerances of all species restricted to the Gulf of Oman were within the 95% bounds of the species present in the Arabian Gulf.

In terms of critical thermal minima ($CT_{min}$), all species, regardless of origin, tolerated the minimum winter temperature of the southern Arabian Gulf at 17.3 °C. Among individuals sampled from the Gulf of Oman population, *E. pulcher* had the greatest tolerance to cold temperatures ($CT_{min} = 11.3 \pm 0.1$ °C), while *E. ventermaculus* had the poorest tolerance ($13.3 \pm 0.1$ °C). The cold-tolerance of *E. ventermaculus* in the Arabian Gulf ($12.3 \pm 0.06$ °C SE) exceeded its Gulf of Oman counterpart ($13.3 \pm 0.10$ °C SE) (Supplementary Table 4), which provides evidence from a second population for intraspecific differences in thermal tolerances across the two locations. Although there were again species-specific differences in the critical thermal minimum, mean cold tolerances of all Gulf of Oman species also fell within the 95% credible bounds of the species present in the Arabian Gulf (Fig. 3a).

**Prey ingestion.** To further examine the potential drivers of cryptobenthic community structure, we quantified prey ingestion in the two locations using gut content DNA metabarcoding[66] across 88 individuals belonging to six species (*C. anomolus*, *E. pulcher*, and *E. ventermaculus* [Arabian Gulf and Gulf of Oman populations]; *Antennablennius adenensis*, *E. guttata*, and *H. vulgaris* [Gulf of Oman only]). We targeted the cytochrome *c* oxidase subunit I (COI) gene region with primers that preferentially amplify metazoans and the 23S rRNA gene region with primers designed to amplify algae. Across all examined fishes, COI metabarcoding yielded a total of 547 unique operational taxonomic units (OTUs), while 23S metabarcoding yielded 3009 unique exact sequence variants (ESVs). Bipartite dietary network trees and modularity analyses for the COI marker showed strong separations between the Arabian Gulf and Gulf of Oman populations (Fig. 4). The COI network contained five distinct modules (modularity = 0.472), with 92.3% of individuals from the Arabian Gulf distributed across two modules. Module V contained seven

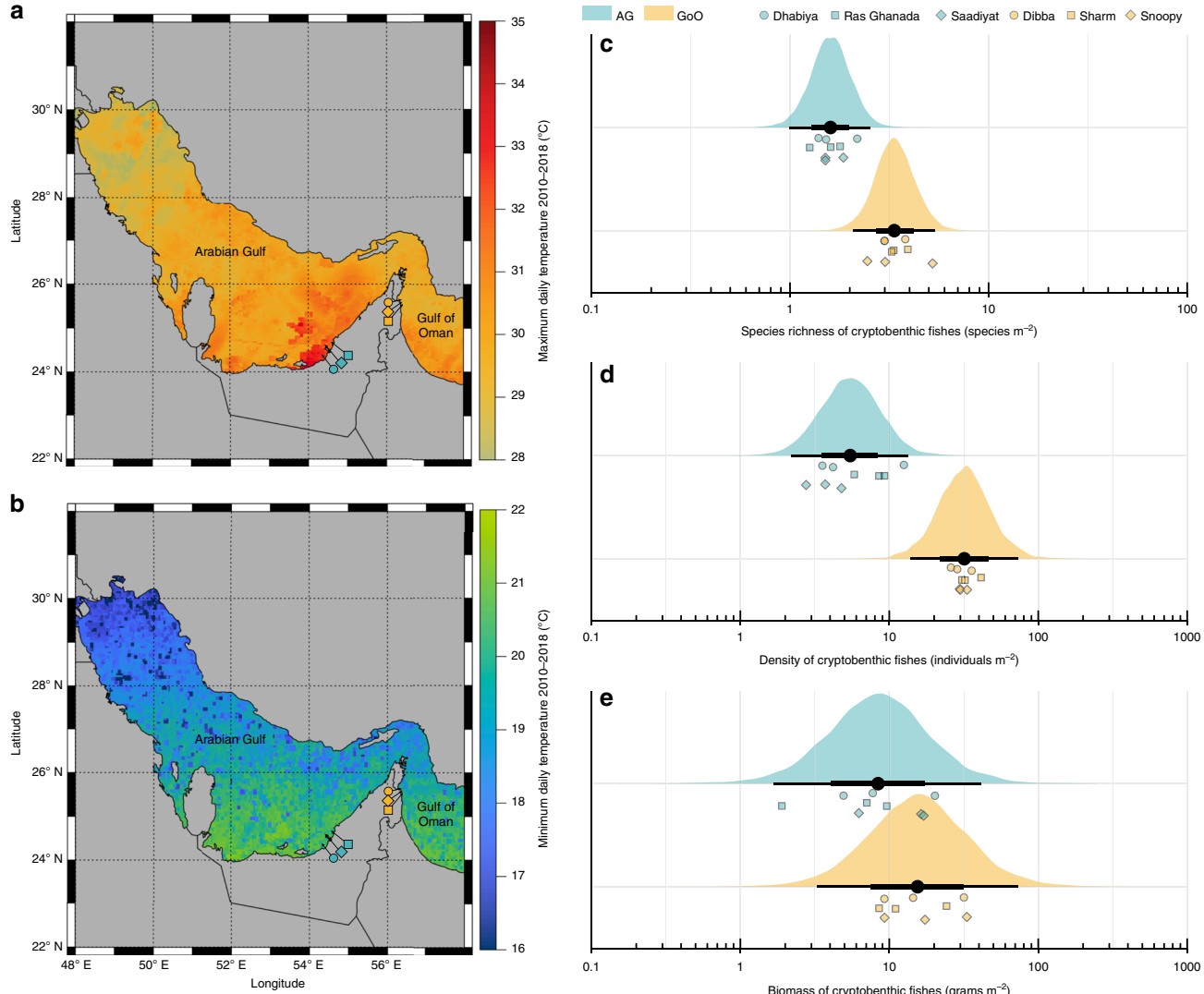

**Fig. 1 Map of the study system and community structure of cryptobenthic reef fish communities in the Arabian Gulf (AG) and Gulf of Oman (GoO).** **a**, **b** Maximum and minimum daily temperature estimates between 2010 and 2018, with the study sites indicated. **c**−**e** Species richness, abundance, and biomass of cryptobenthic fish communities. Density curves and black caterpillar plots (50 and 95% credible intervals) represent posterior predictions, while circles, squares, and diamonds represent raw values from the respective sites in each location, jittered on the y-axis.

out of ten individuals of *C. anomolus* from the Arabian Gulf, eight out of nine individuals of *E. ventermaculus* from the Arabian Gulf, and one *E. guttata* from the Gulf of Oman. The remaining individuals of *C. anomolus* and *E. ventermaculus* from the Arabian Gulf clustered with *E. pulcher* from the Arabian Gulf (five out of seven), four Gulf of Oman individuals of *C. anomolus*, and a single *H. vulgaris* in module II (Fig. 4a, b). The 23S marker also revealed five modules (modularity = 0.359) but showed an even stronger regional separation. All individuals from the Arabian Gulf (except for one individual of *C. anomolus*) were united in a single module (module III), which contained no Gulf of Oman individuals (Fig. 4c, d). While some species separated into distinct modules, location-specific differences superseded taxonomic boundaries. With the exception of *C. anomolus*, species occurring in both locations showed strong dietary differences, while broadly overlapping with other species in the Gulf of Oman.

Prey diversity rarefaction curves in the Gulf of Oman showed that *E. pulcher*, a purportedly herbivorous species[67], ingested the widest variety of animal prey (COI marker), followed by *E. ventermaculus* (Supplementary Fig. 3). For both species, Gulf of

Oman populations consumed a higher diversity of prey items than Arabian Gulf populations. Only *C. anomolus* showed no clear difference in extrapolated values (although diversity was higher for Gulf of Oman populations for the interpolated value). For algal prey items (23S marker), prey diversity was again higher in Gulf of Oman populations of *E. pulcher* and *E. ventermaculus*, while the opposite was evident for *C. anomolus*. Overall, Gulf of Oman populations of *E. ventermaculus* ingested the highest autotroph prey diversity, followed by Arabian Gulf populations of *C. anomolus*.

**Energetic consequences at the organismal- and ecosystem-scale.** We further examined the potential organismal and ecosystem-wide energetic consequences of thermal regimes, organismal responses, and resource availability between the two locations by first assessing length−weight relationships of three co-occurring species, and then by modeling individual-based growth and mortality to estimate community-wide biomass cycling. We employed Bayesian linear models to test the effects of total length (*TL*) and *Location* on *Weight*, which showed clear effects of

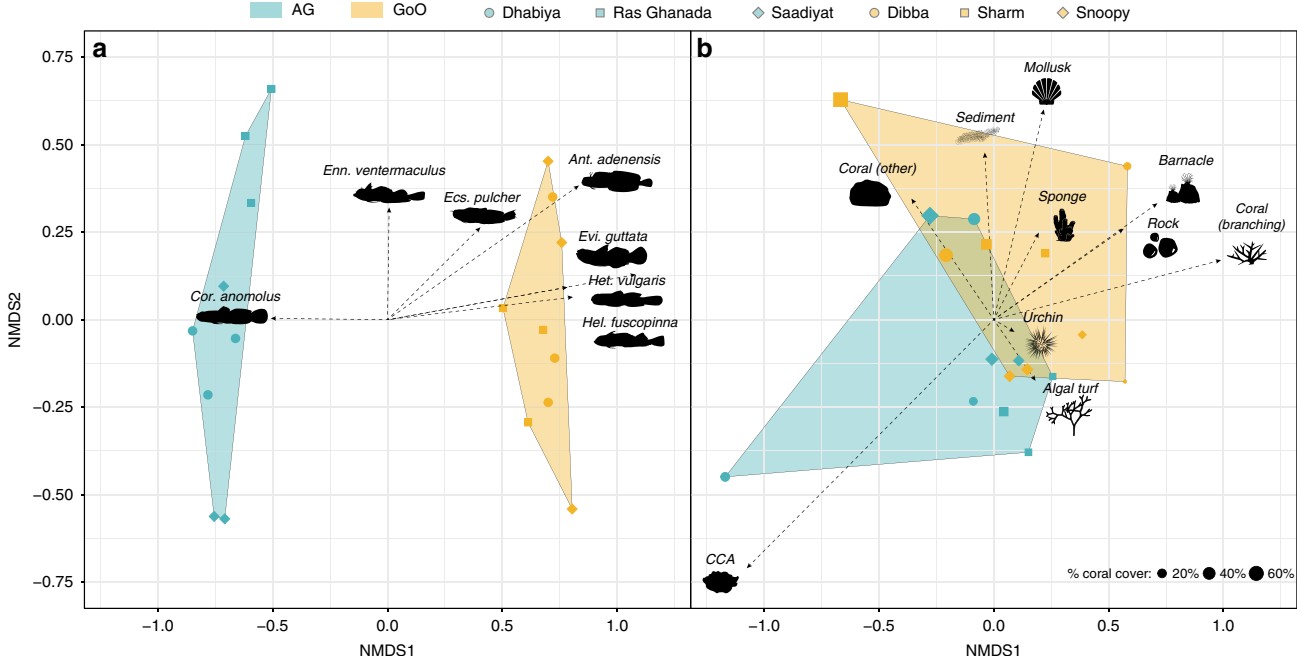

**Fig. 2 Community composition of cryptobenthic reef fishes and benthic functional/taxonomic groups in the Arabian Gulf (AG) and Gulf of Oman (GoO). a** Biplot of a nonmetric multidimensional scaling (nMDS) ordination on cryptobenthic fish communities, with the arrows indicating the position and strength of the seven most important species. **b** Biplot of an nMDS on benthic functional groups, with the influence of all groups indicated with arrows. Convex hull polygons delineate locations. Each point represents a sample station at a particular site, with the shape size in (**b**) scaled by percent live coral cover. CCA = crustose coralline algae.

*Location* across all three species, with Gulf of Oman populations consistently having higher weights for a given body length (*E. ventermaculus*: *Gulf of Oman*: $\beta = 0.16$ [0.13, 0.19], *E. pulcher*: *Gulf of Oman*: $\beta = 0.19$ [0.14, 0.25]), and *C. anomolus*: *Gulf of Oman*: $\beta = 0.15$ [0.09, 0.21] (Fig. 5). Specifically, at each species' mean total length, our model predictions show that individuals of *E. ventermaculus*, *E. pulcher*, and *C. anomolus* were 67.2%, 62.2%, and 10.0% heavier in the Gulf of Oman, respectively. Notably, empirical values for the largest individuals of *C. anomolus* from the Arabian Gulf were consistently below the model fit, suggesting worse body conditions than predicted by the model and substantially worse body conditions than Gulf of Oman individuals of comparable size (Fig. 5b). In contrast, no clear differences emerged between the abundances of the three species' populations across locations (effect size uncertainties intersected zero), although *E. ventermaculus* (*Gulf of Oman*: $\beta = 0.89$ [−1.08, 2.86]) and *E. pulcher* (*Gulf of Oman*: $\beta = 3.46$ [−0.42, 9.93]) showed a trend toward lower abundances in the Arabian Gulf, while *C. anomolus* exhibited the opposite trend (*Gulf of Oman*: $\beta = -0.94$ [−3.82, 1.69]).

Finally, modeling individual-based growth and mortality for cryptobenthic fish communities at each site revealed strong differences between the Arabian Gulf and Gulf of Oman in the ecological dynamics that underpin coral reef ecosystem functioning (Fig. 6). Biomass production was almost one order of magnitude lower on reefs in the Arabian Gulf ($0.038 \pm 0.014$ g d$^{-1}$ m$^{-2}$) compared to the Gulf of Oman ($0.231 \pm 0.025$ [mean ± SE] g d$^{-1}$ m$^{-2}$), while consumed biomass production was more than five times lower ($0.007 \pm 0.001$ g d$^{-1}$ m$^{-2}$ vs. $0.039 \pm 0.015$). Turnover was also lower in the Arabian Gulf ($0.006 \pm 0.005\%$ d$^{-1}$) compared to the Gulf of Oman ($0.017 \pm 0.005\%$ d$^{-1}$). Therefore, coral reefs in the two locations exhibit contrasting productivity dynamics at various levels of organization. In the Arabian Gulf, individual fishes accumulate less body mass per millimeter of body length and collectively, cryptobenthic communities produce, provide, and replenish consumer biomass at much lower rates than Gulf of Oman communities.

## Discussion

As rapid environmental change sweeps across Earth's ecosystems, understanding the processes that underpin local community structure and ecosystem functioning is critical. Here, we show that cryptobenthic fishes on the world's most environmentally extreme reefs in the southeastern Arabian Gulf have reduced diversity, abundance, and body condition compared to reefs with more moderate temperatures in the nearby Gulf of Oman, despite similarities in live coral cover and benthic community structure. While we found some evidence for intraspecific thermal plasticity, which may enable survival in Arabian Gulf conditions, species-specific temperature tolerances are not the main driver of species presence/absence in the Arabian Gulf. Rather, poor body condition in Arabian Gulf populations alongside intraspecific differences in the diversity and composition of ingested prey items across the two locations indicate that the Arabian Gulf represents an energetically challenging environment that prevents the persistence of many small-bodied ectotherms. This has cascading consequences for ecosystem-scale energy and nutrient fluxes, as even conservative estimates of cryptobenthic reef fish productivity in the Arabian Gulf are an order of magnitude lower than the Gulf of Oman. Our results indicate that cryptobenthic reef fish assemblages on future coral reefs may be shaped by species-specific individual energy deficits that decrease the rate of biomass production, transfer, and renewal through small vertebrate consumers, thereby eroding a cardinal component of heterotrophic coral reef productivity[53].

As the smallest and shortest-lived marine vertebrates, responses of cryptobenthic fishes to extreme temperatures should be easy to trace[49]. Yet, critical thermal tolerances of all tested species

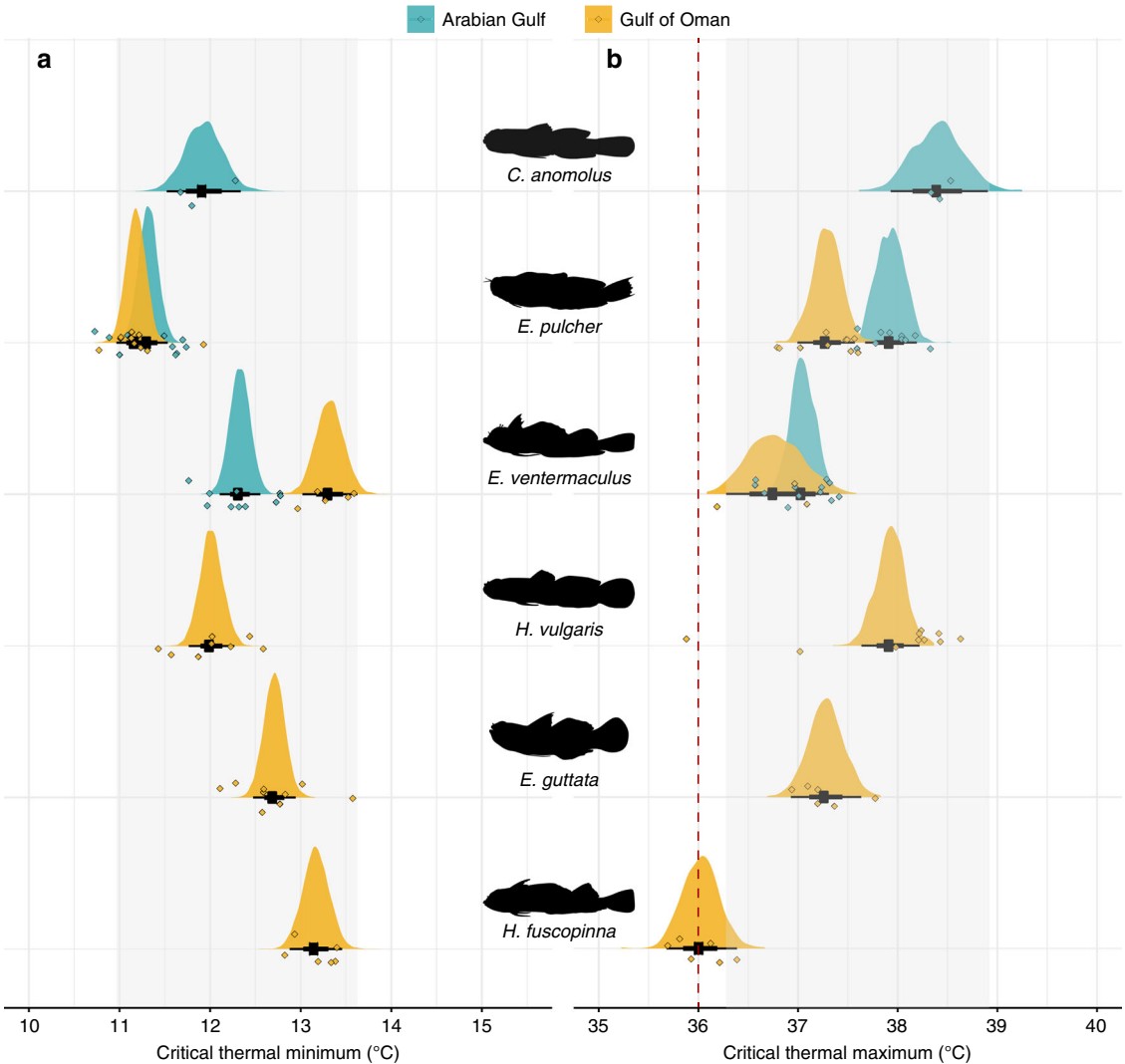

**Fig. 3 Critical thermal tolerance limits of cryptobenthic fish species from the Arabian Gulf and Gulf of Oman. a** Average critical thermal minima and **b** average critical thermal maxima. The red dashed line (36 °C) indicates the maximum temperature recorded in the Arabian Gulf. Density curves and black caterpillar plots (50 and 95% credible intervals) represent fitted values from Bayesian linear models. Diamonds represent raw values, jittered on the *y*-axis. Gray boxes delineate the range of the 95% credible intervals obtained for the three species present in the Arabian Gulf.

from both locations were equal to or greater than the extreme maximum summer temperatures of the southeastern Arabian Gulf[41,44,68]. The high intrinsic temperature tolerance of species from the thermally moderate Gulf of Oman aligns with previous results of high, short-term critical thermal tolerances in cryptobenthics[41]. Furthermore, provided that self-recruitment is high for cryptobenthic fishes[53], swift generational turnover in cryptobenthic fishes could facilitate transgenerational thermal plasticity and increased thermal tolerance[53,57]. Collectively, this should have permitted their colonization and persistence in the geologically young southeastern Arabian Gulf[63] since no hard biogeographic boundary exists between the Gulf of Oman in the Arabian Gulf[64]. Indeed, 26 out of 29 (89.7%) cryptobenthic fish species from the Gulf of Oman that were absent from the southeastern Arabian Gulf (where temperatures are extremely high in the summer, but moderate in the winter) have been recorded in the cooler Arabian Gulf regions of Saudi Arabia and Kuwait (Supplementary Table 2)[65,69,70]. Thus, neither thermal tolerances to short-term temperature extremes nor biogeographic history are likely to drive the observed depauperate cryptobenthic communities on Earth's hottest coral reefs. While it is possible that organismal tolerances to other environmental factors, such as

elevated salinity, may play a part in the observed patterns, temperature is generally considered to be the primary environmental force that shapes Arabian Gulf communities[61,62,71].

In the absence of a direct lethal effect of temperature or other environmental factors, our results suggest that more nuanced forces based on shifting thermal tolerances, their sublethal physiological effects, and distinct prey resources across the two locations shape the observed assemblages. Shifting tolerance in response to increasing temperatures can incur substantial energetic costs related to metabolic and molecular processes[36,46,72]. These costs (necessitated by either extreme temperatures or overall environmental variability) are evident in the lower mass per unit body length of Arabian Gulf populations in the three examined species. At the time of sampling (end of spring), two out of three species were more than 60% lighter at their mean body length in the Arabian Gulf, suggesting substantial deficits in condition[73–75]. Since temperatures are generally comparable between the two locations in the spring (Supplementary Fig. 2), and spring is typically when animals accrue body mass between seasonal extremes, the poor body condition found in Arabian Gulf populations may be a consequence of carryover effects arising from transgenerational processes that enable survival

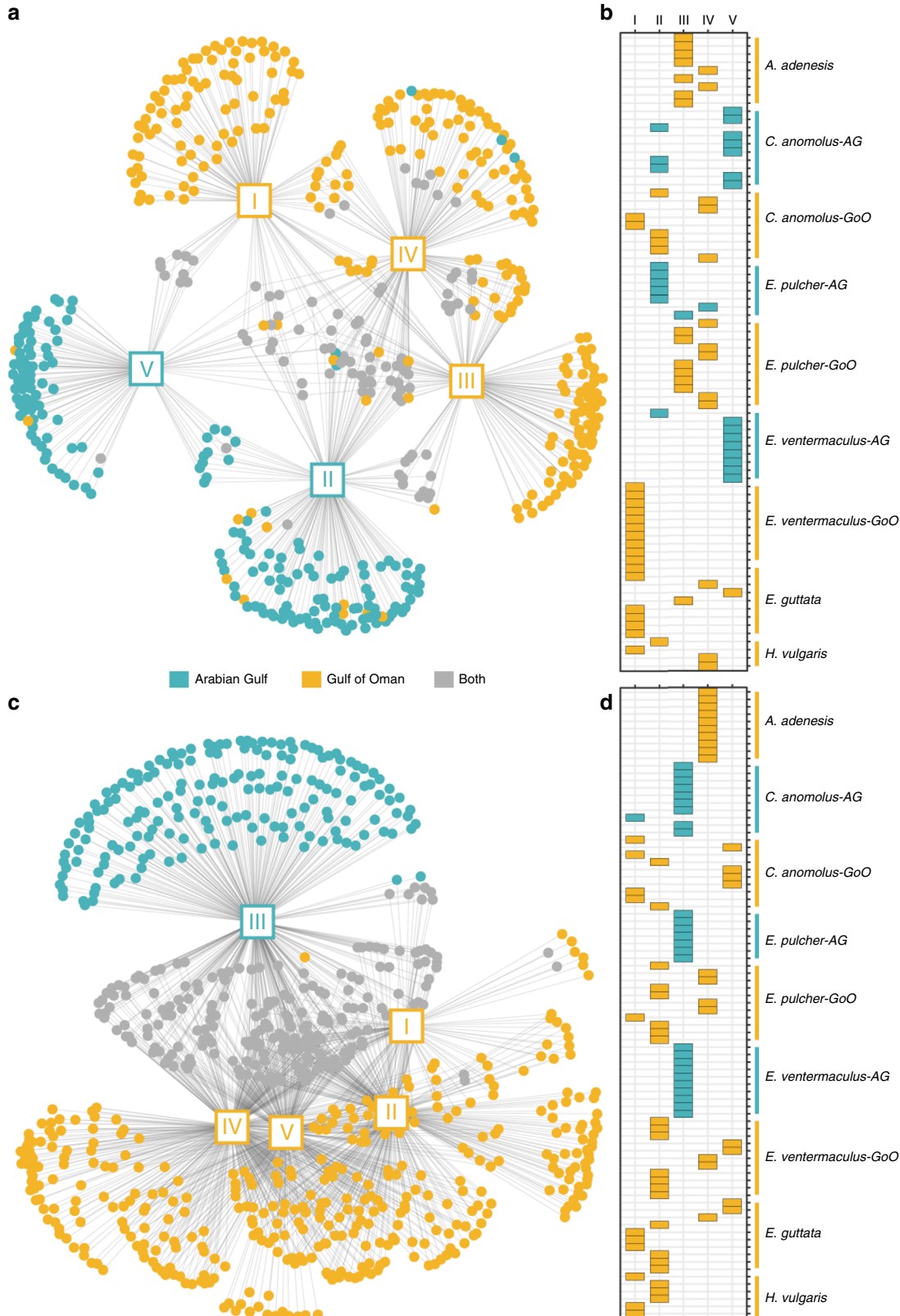

**Fig. 4 Patterns of prey ingestion by cryptobenthic fishes in the Arabian Gulf (AG) and Gulf of Oman (GoO).** Diet network trees and modularity mosaics show differences in ingested prey items and individual-based module membership for COI (**a**, **b**) and 23S (**c**, **d**) markers. **a**, **c** Squares with roman numerals represent the recovered modules as nodes in the network tree, while dots represent unique prey items. Blue dots are OTUs (COI) or ESVs (23S) found only in individuals from the Arabian Gulf, gold symbols are from the Gulf of Oman individuals, and gray symbols represent prey items found in individuals from both locations. **b**, **d** Results of the modularity analysis with modules (I−V) as columns and individuals within each species as rows. Colored tiles indicate membership in a given module.

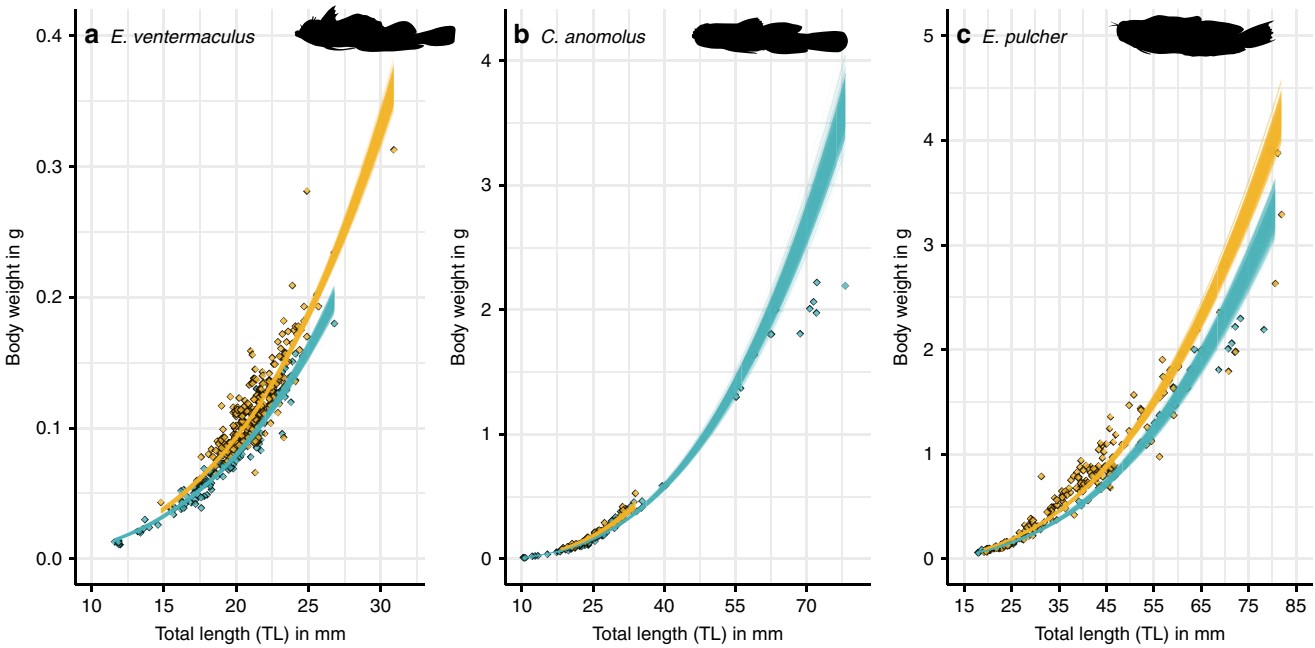

**Fig. 5 Relationships between total length (TL) and body weight in cryptobenthic fish populations from the Arabian Gulf (blue) and Gulf of Oman (gold).** Each line represents a fitted draw from 500 iterations based on the posterior parameters from a Bayesian model regressing length against weight (thus showing model fit uncertainty). Diamonds represent raw values for individual fishes. Panels **a**–**c** represent *E. ventermaculus, C. anomolus* and *E. pulcher,* respectively.

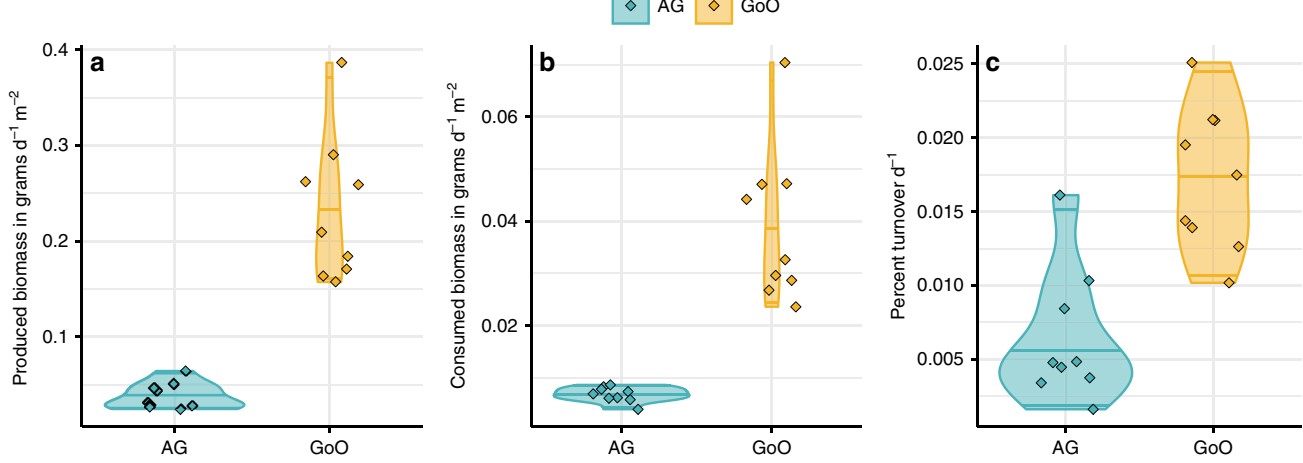

**Fig. 6 Estimated biomass production, consumption, and turnover in cryptobenthic fish assemblages in the Arabian Gulf (AG) and Gulf of Oman (GoO).** **a** Produced biomass (grams of fish tissue grown per day and m²). **b** Consumed biomass (grams of fish tissue perished per day and m²). **c** Percent turnover (renewal of produced and consumed biomass per day). Violin plots and lines represent medians and variance estimates (95% confidence intervals) for the three metrics across the two locations. Diamonds represent values for each sampled cryptobenthic reef fish community across the six sites.

(through increased thermal tolerance), but significantly hamper physiological condition (e.g., upregulation of metabolic rates, liver inflammation[43,46]).

The energetic burden imposed by shifts in thermal tolerance may be exacerbated by fundamentally different prey resources and reduced prey diversity; indeed, gut content metabarcoding revealed a different and narrower range of prey resources ingested by individuals from the Arabian Gulf. Shifts in prey composition often necessitate changes in digestive efficiency that may require radical physiological or morphological adjustments, ultimately affecting species' energy budgets[76,77]. Furthermore, a lower diversity of prey items can reduce individual and population persistence[78,79]. Naturally, energetic challenges will be even greater if prey in the Arabian Gulf have less favorable nutritional profiles or energy

densities[80]. While we did not investigate differences in diet quality (i.e., nutrient content, energetic yield) or quantities of prey across locations, large reef fish species in the Arabian Gulf ingest unusual diets dominated by nutritiously poor benthic invertebrates[81]. Collectively, our findings suggest that shifts in thermal tolerances in response to environmental extremes, and their associated energetic costs, may not be a viable strategy for small, tropical ectotherms if environments are resource-limited, either in quality, quantity or both. Thus, although the adjustment of thermal tolerances following environmental change has been shown to enable survival in controlled laboratory conditions via several microevolutionary processes[36,46], their role may be limited in the wild, where animals continuously engage in costly activities such as foraging or escaping predators[73].

In turn, the case of the goby *C. anomolus* emphasizes the potential importance of deeply rooted evolutionary processes in permitting persistence in extreme environments. *C. anomolus* was the only species to show weakly distinct prey composition between locations and higher autotroph prey richness in the Arabian Gulf, and had a higher abundance and larger body size in the Arabian Gulf as compared to the Gulf of Oman. Furthermore, compared to *E. pulcher* and *E. ventermaculus*, differences in body condition between locations were weak for *C. anomolus*. As opposed to most dominant cryptobenthic genera in the Arabian Gulf and Gulf of Oman (e.g. *Ecsenius, Eviota, Enneapterygius*, etc.), the goby genus *Coryogalops* belongs to a clade that contains many non-reef-associated species from comparatively extreme habitats[82,83]. For example, *Coryogalops* often inhabit tidepools and other shallow environments exposed to fluctuating temperatures and salinity where they rely on a sedentary lifestyle with low energetic costs[84,85]. Thus, the persistence of thriving *C. anomolus* populations in the southeastern Arabian Gulf may reflect deeper evolutionary rooting in extreme environments, which permits this species to satisfy its energetic demands with prey resources present in both locations.

Our results indicate that species-specific capacities to cope with the energetic costs of inhabiting extreme environments, rather than the direct effects of temperature per se or its effect on benthic community structure (cf. ref. [68]), underpin the limited diversity and abundance of cryptobenthic fishes on these extreme reefs. For cryptobenthics, which already exhibit high energetic demands per gram of body mass[49], augmented energetic costs likely represent a crucial challenge. Along with environmentally driven differences in prey composition and diversity (and possible reductions in nutritional value or energetic densities), this "energetic double jeopardy" may represent an insurmountable obstacle for many cryptobenthic species. Further decreases in body size (a universal physiological response to warmer temperatures[15,74]) might simply be impossible for many cryptobenthic reef fishes that are already at or near the physical minimum body size for vertebrates[49,54,86]. Therefore, our findings from small-bodied tropical ectotherms in a natural setting suggest that more extreme environmental conditions, as predicted for future reefs due to climate change, may have severe consequences on organismal performance[87,88], with cascading effects on species persistence and community assembly[89].

The organismal drivers of community assembly in the southeastern Arabian Gulf create a sobering perspective on coral reef ecosystem functioning in a more extreme, rapidly warming ocean. Coral reefs are highly productive marine ecosystems[90] that are sustained through a variety of energetic pathways[91–94]. Among these pathways, benthic productivity[95] and its assimilation and transfer through cryptobenthic reef fishes represents an important bottom-up flux of energy and nutrients to higher trophic levels[53]. The differences in biomass production, transfer, and turnover between cryptobenthic fish communities in the Arabian Gulf and Gulf of Oman suggest that the role of cryptobenthics as vectors of energy and nutrients to larger consumers may be stymied in extreme environments. In fact, yearly productivity estimates for cryptobenthic fishes in the Arabian Gulf may be even lower than our model suggests due to the decreased individual-level production of body mass per unit body size and the influence of seasonality effects on growth[96]. Yet, neither environmental limits on the growing season, nor decreased individual mass per unit body size were considered in the model.

The Gulf of Oman reefs included in this study may be particularly productive environments due to seasonal upwelling[97], and indeed, our estimates of cryptobenthic productivity exceeded estimates for a degraded but species-rich reef on the Australian Great Barrier Reef (GBR) (2.31 vs. 0.64 kg ha$^{-1}$ d$^{-1}$)[98]. In contrast, even the optimistic estimate of 0.38 g ha$^{-1}$ d$^{-1}$ for the Arabian Gulf compared poorly with the same degraded GBR-reef. Notably, the

study site on the GBR had undergone a sequence of severe disturbances[98], yet it retained a diverse assemblage of cryptobenthic fish species that were likely able to satisfy their energetic demands due to benign temperature profiles[26]. At the time of our survey, reefs in the Arabian Gulf had undergone extensive bleaching in previous years[99–102], which may have negatively affected the diversity and abundance of cryptobenthic fishes compared to the less disturbed reefs in the Gulf of Oman[24,103,104]. Furthermore, larger-scale structural differences between reef outcrops in the Gulf of Oman and the Arabian Gulf may affect our community-wide estimates. However, the lack of difference in benthic community structure observed between regions suggests that benthic structure was at least not a primary driver of the observed patterns. Although the loss of some specialist cryptobenthic species has been reported after substantial live coral cover loss[99,105], previous studies have not detected substantial short-term changes in either small reef fish richness and abundance or in overarching ecosystem productivity[27,29,31,56,99].

Our results showcase an imminent threat to cryptobenthic reef fishes and their role for coral reef functioning: many of the world's smallest marine ectotherms may struggle to compensate for increasing costs of growth and homeostasis as they adapt to more extreme environmental regimes. As a consequence, small consumer productivity, energy transfer, and replenishment of biomass at the bottom of the fish food chain may decrease under climate change[15]. Analogous to cryptobenthics, the Arabian Gulf harbors less diverse and abundant communities of large reef fishes compared to nearby locations with more moderate temperatures[71,106]. It remains unresolved whether these patterns are driven by similar mechanisms as proposed herein (e.g., an energetic filtering effect on large fish species) or relate to decreased productivity at lower trophic levels. Yet, in light of the hypothesized importance of small vertebrate consumers in global food webs[107] and the unique ecological role of cryptobenthics in coral reef trophic dynamics[53], the effects of elevated temperature on cryptobenthic fish assemblages may considerably reduce ecosystem functioning on future coral reefs.

## Methods

**Field sampling.** We studied cryptobenthic fish communities in two locations that dramatically differ in their annual temperature profiles. Temperatures in the Arabian Gulf (Dhabiya: 24.36383°, 54.10121°; Ras Ghanada: 24.84743°, 54.69235°; Saadiyat: 24.65771°, 54.48691°) are extremely hot, with summer maximum temperatures reaching up to or above 36 °C, while winter minimum temperatures fall to 16 °C. In contrast, temperatures in the Gulf of Oman (Dibba Rock: 25.55378°, 56.35694°; Sharm Rock: 25.48229°, 56.36695°; Snoopy Rock: 25.49210°, 56.36401°) lie within more typical coral reef temperature profiles throughout the year, ranging from 32 to 22 °C[64]. Notably, the sampled reefs in the Arabian Gulf are at the extreme end of high maximum summer temperatures, while being relatively benign concerning the low winter temperatures in the rest Arabian Gulf (Fig. 1a, b). At the time of sampling, temperatures between the two locations were between 27 and 29 °C at both locations. All in situ temperature data (Supplementary Fig. 2) were obtained from HOBO data loggers deployed on the substrate (4−6 m depth) at the respective sites (cf. ref. [60]).

In April and May of 2018, we sampled six reefs (hereafter *site*) in the southeastern Arabian Gulf and northwestern Gulf of Oman (three sites per location). At each site, we sampled three distinct reef outcrops for cryptobenthic reef fishes using enclosed clove-oil stations[50,108], covering an average of 4.63 ± 0.38 and 4.73 ± 0.16 m$^2$ in the Arabian Gulf and Gulf of Oman, respectively, for a total of 18 community samples. Since our sampling was not replicated temporally, we cannot exclude the possibility of annual changes in cryptobenthic communities in the Arabian Gulf. Nevertheless, the lack of records for many of the species found in the Gulf of Oman in the southeastern Arabian Gulf indicates that the depauperate nature of cryptobenthic assemblages in this region is not a function of our sampling at a single point in time. For each station, we covered a reef outcrop with a fine-mesh, bell-shaped net (2.74 m in diameter), weighted by a chain on the bottom. We then covered the same area with an impermeable bell-shaped tarpaulin, also weighted by a chain on the bottom. Then, 3−4 divers inoculated the area under the net with two liters of clove-oil:ethanol solution (1:5) using collapsible spray bottles (clove bud oil: Jedwards International, Inc., Braintree, MA, USA). Upon emptying the entire solution and a short wait period to allow the clove oil to disperse and take effect (approximately 2−3 mins), we removed the tarpaulin and gently peeled

back the net while collecting all fishes found within the inoculated area with tweezers. We searched the entire area, including inside caves and crevices until 5 min passed without a single diver collecting any additional fishes. We placed all fishes into Ziplock bags, brought them to the surface, euthanized them with a clove-oil overdose, and immediately placed them into an ice-water slurry until processing and preservation. At the end of each day, all specimens were brought to the laboratory at NYUAD or the Radisson Blu hotel in Fujairah. To quantify benthic community structure, we used a haphazardly placed 20 × 20 cm PVC-quadrat to frame and take five photographs of the benthos at each sampled outcrop.

In addition to the quantitative samples obtained from the clove-oil stations, we collected cryptobenthic fish individuals for thermal tolerance trials using roving diver collections. Specifically, two divers, each equipped with spray bottles of clove-oil:ethanol solution, a dipnet, and Ziplock bags, searched the reef for cryptobenthic fishes across three species in the Arabian Gulf (*C. anomolus*, *E. pulcher*, and *E. ventermaculus*) and six species in the Gulf of Oman (*C. anomolus*, *E. pulcher*, and *E. ventermaculus* plus *E. guttata*, *H. fuscopinna*, and *H. vulgaris*). Upon locating an individual or identifying a suitable microhabitat in which a fish was suspected, the diver applied the clove-oil solution until the fish showed signs of anesthesia. At the earliest opportunity, we caught the fish with a dipnet and placed it into a ziplock bag. Upon completion of the dive, all fishes were placed in small holding tanks equipped with air stones and periodically replenished with fresh seawater. Upon completion of all collections, fishes were brought to the seawater laboratory facilities at NYUAD. All roving diver collections were performed at Dhabiya Reef (Arabian Gulf) and Snoopy Rock (Gulf of Oman).

**Laboratory processing**. For samples obtained from the enclosed clove-oil stations, we followed an established protocol that involved photographing, identifying, recording, measuring, weighing and preserving each specimen[50]. To photograph the fishes, we placed each individual in a small photo tank and used a Nikon D300 DSLR camera with an AF-S Micro Nikkor 60 mm macro lens (f/2.8 G ED; Nikon Inc., Melville, NY, USA) against a black or white background. We measured each individual to the nearest 0.1 mm using digital calipers and weighed the individual (wet weight) to the nearest 0.001 g on a precision jewelry scale. We preserved all individuals in 95% ethanol, either separately or in lots with conspecifics. A subset of the samples was shipped to the University of Washington Fish Collection, where they were cataloged, while the rest were retained and archived at NYUAD.

**Benthic photo analysis**. For the benthic photographs, we created a grid with 16 equally spaced points which we superimposed on every photograph. We then categorized the benthos at each of the points into functional groups, including barnacles, bleached corals, crustose coralline algae, dead coral, hydroids, branching, encrusting, foliose, and massive live coral, mollusks, bare rock, soft sediment, sponges, algal turf, and sea urchins. Whenever visual identification was not possible (due to obstruction, shading, or blurriness), we categorized the point as "uni-dentifiable" ($n = 69$ out of 1440). All photographs with the grid superimposed are accessible with the raw data of the paper.

**Critical thermal maximum and minimum trials**. We examined individual temperature tolerances by using critical thermal maximum ($CT_{max}$) and minimum ($CT_{min}$) trials[109]. We transported all fishes caught during roving diver collections to the wet laboratory facilities at NYUAD and housed them for at least 48 h in large holding tanks. Trials took place from the 9th to 13th of May 2018. For the trials, a haphazardly selected subset of individuals was moved from the holding tanks into separate chambers filled with seawater at ambient temperature and salinity. Then, after providing individuals with a 15-min settlement period, we incrementally decreased ($CT_{min}$) or increased ($CT_{max}$) the water temperature within the chambers while keeping all other parameters constant. Specifically, we lowered or increased the temperature by 0.1 °C every minute[109] while keeping all fishes under constant observation. Critical endpoints were classified as loss of equilibrium or uncontrolled swimming without a righting response for 2 s or more[109]. When individuals reached their critical endpoints, they were immediately removed, euthanized with a clove-oil overdose, measured, weighed, and photographed. In total, we processed 60 individuals across six species for $CT_{max}$ trials, and 62 individuals across the same species for $CT_{min}$ trials. Specific sample sizes are provided in the supplementary material (Supplementary Table 5).

**Gut content DNA metabarcoding**. We processed a subset of individuals across six species (*A. adenensis*, *C. anomolus*, *E. pulcher*, *E. guttata*, *E. ventermaculus*, and *H. vulgaris*) for gut content DNA metabarcoding at the University of Washington. We haphazardly selected ten, ten, and seven (due to limited sample availability) individuals of *C. anomolus*, *E. ventermaculus*, and *E. pulcher*, respectively, from the Arabian Gulf, and ten individuals each (with the exception of *E. pulcher*, for which we selected 11 individuals) of *C. anomolus*, *E. ventermaculus*, *A. adenensis*, *E. guttata*, and *H. vulgaris* from the Gulf of Oman. Then, under sterile conditions, we dissected out the entire alimentary tract and removed all other organs (e.g. liver, gonads) under a Zeiss V20 SteREO dissecting microscope using micro-surgery tools. We placed the entire gut into an extraction tube and performed DNA

extractions with a DNeasy PowerSoil Pro DNA Isolation Kit (Qiagen, Hilden, Germany). We stored all DNA extracts at 4 °C until further processing.

All DNA samples were sent to Jonah Ventures (Boulder, Colorado, USA) for two-step PCRs, library preparation, and sequencing. We targeted two universal gene regions: the mitochondrial COI for metabarcoding metazoan biodiversity and the chloroplast 23S rRNA for metabarcoding algae. For the COI gene, we selected the m1COIintF forward primer[110] and jgHCO2198 reverse primer[111]. For the 23S gene, we selected the p23SrV_f1 and Diam23Sr1 23S primers[112–114]. All COI and 23S primers contained a 5′ adaptor sequence to facilitate indexing and sequencing. The PCR reactions for both COI and 23S genes were run at a volume of 25 μl according to the Promega PCR Master Mix guidelines (Promega, Madison, Wisconsin, USA): 12.5 μl Master Mix, 0.5 μM of each primer, 1 μl gDNA, and 10.5 μl DNase/Rnase-free water. For COI, PCR amplification was run with the following conditions: initial denaturation at 94 °C for 2 min, followed by 45 cycles of 15 s at 94 °C, 30 s at 50 °C, and 1 min at 72 °C, then a final elongation at 72 °C for 10 min. For 23S, DNA was PCR-amplified under the following conditions: initial denaturation at 94 °C for 3 min, followed by 40 cycles of 30 s at 94 °C, 45 s at 55 °C, and 1 min at 72 °C, then a final elongation at 72 °C for 10 min. After PCR amplification, each reaction was visually inspected with a 2% agarose gel to ensure successful amplification and determine amplicon size.

All remaining library preparation and sequencing protocols apply to both the COI and 23S markers. Clean-ups were performed by incubating amplicons with Exo1/SAP for 30 min at 37 °C, followed by inactivation at 95 °C for 5 min, then the products were stored at −20 °C. Next, a second indexing PCR was performed to bind a unique 12-nucleotide index sequence. The PCR reaction included Promega Master mix, 0.5 μM of each primer, and 2 μl of template DNA. The PCR was performed with the following conditions: initial denaturation at 95 °C for 3 min, followed by eight cycles of 95 °C for 30 s, 55 °C for 30 s, and 72 °C for 30 s. Each reaction was visually inspected with a 2% agarose gel to ensure successful amplification.

A volume of 25 μl of each indexed amplicon was cleaned and normalized with the SequalPrep Normalization Kit (Life Technologies, Carlsbad, California, USA) according to the manufacturer's protocol. For sample pooling, 5 μl of each sample was added together. Finally, library pools were sent to the Genohub service provider (Austin, Texas, USA). Prior to sequencing, quality control measures were performed, including bead cleaning with Agencourt AMPure XP beads (Beckman Coulter, Brea, California, USA) to remove <200 bp amplicons, sample quantification with a Qubit Fluorometer (Invitrogen, Carlsbad, California, USA), and amplicon average size analysis with an Agilent TapeStation 4200 (Agilent, Santa Clara, California, USA). Finally, sequencing was performed on an Illumina HiSeq using the HiSeq Rapid SBS Kit v2, 500-cycles (Illumina, San Diego, California, USA).

**Sequence bioinformatics**. For the COI sequences, a joint QIIME[115] and UPARSE[116] pipeline was employed for bioinformatic processing. Sequences were demultiplexed and initial quality filtering was performed with QIIME v1.9.1. Primer sequences were trimmed with Cutadapt v1.18 [117], then forward and reverse reads were pair-end merged with USEARCH v11.0.667 [118]. Quality filtering was then performed in accordance with the UPARSE pipeline. Sequences were clustered into OTUs at 99% similarity, and the OTU table was generated by mapping quality-filtered reads back to the OTU seeds. Taxonomy was assigned to OTUs by recording the top basic local alignment search tool (BLASTn[119]) hit when query coverage and percent identity exceeded 95% and 80%, respectively. GenBank was used as the reference database. When OTU taxonomic assignments did not meet these criteria, taxonomy was removed and recorded as "NA." Finally, we removed all self-hits from the OTU-dataset, which we identified by matching the highest sequence reads of each species to its individuals, as well as unambiguous (>97% identity match) assignments to species not found in the geographic region (specifically *Oncorhynchus nerka*).

For the 23S raw sequences, raw sequences were processed with the JAMP pipeline (https://github.com/VascoElbrecht/JAMP). After demultiplexing, forward and reverse reads were pair-end merged with USEARCH v11.0.667 [118]. Primers were trimmed from both ends using Cutadapt v1.18 [117], and quality filtering was conducted with expected error filtering, as implemented through USEARCH[120]. Reads affected by sequencing and PCR error were removed using the UNOISE algorithm[121]. Exact sequence variants were then compiled into an ESV table, which included read counts for each sample. Taxonomy was assigned to each ESV by mapping them against a 23S database from Silva[122], specifying zero deviations to ensure mapping accuracy. Consensus taxonomy was generated from the hit tables, first considering 100% matches, then decreasing by 1% until hits were available for each ESV. Taxonomy that was present in at least 90% of the hits was reported; otherwise, an "NA" was assigned when several different taxa matched the ESV. For error reduction due to misidentified taxa, the bracket was increased to 2% when matches of 97% and higher were present, but no family-level or lower taxonomy was assigned.

**Data analyses and modeling**. To analyze the community variables, we first calculated the surface area (SA) for each sampled outcrop from the curved surface length (CSL) by deriving the sampled outcrop's radius $r$ ($r = 2 \times CSL/2\pi$), then computing available surface area under the assumption that outcrops are hemispherical constructs ($SA = 4\pi r^2/2$). We calculated the sum of individuals, species,

and their respective body weight for each station to obtain abundance, diversity, and biomass estimates, which we converted to density estimates by dividing them by the sampled surface area. Using these estimates, we performed three Bayesian hierarchical models, each on the natural logarithm of the response variables (species density, individual density, and biomass per $m^2$). Models were specified to include the fixed effect of *Location* (*Arabian Gulf* vs. *Gulf of Oman*) and the random effect of *Site* (*Dhabiya, Ras Ghanada, Saadiyat, Dibba Rock, Sharm Rock, Snoopy Rock*) and were run with a Gaussian error distribution. For each model, we ran four chains with 4000 post burn-in samples, and we validated chain convergence visually. We used the default, noninformative priors set by the *brm* function in the *brms* R package[123]. Then, we used the model parameters to predict distributions based on 1000 draws from the posterior and plotted the distributions, their mean and confidence bands, and the raw data for each site to evaluate model fit.

To examine cryptobenthic fish community composition across the two locations, we created a species-by-sample matrix indicating the abundance of each species in a given sample. We then performed an nMDS ordination with the Bray−Curtis dissimilarity matrix of the square-root transformed data in two dimensions (stress = 0.101). We performed a permutational analysis of variance (PERMANOVA) on the same distance matrix (using 999 permutations) and extracted the most influential species using the similarity of percentages (SIMPER) routine. We constructed convex hull polygons for the two locations (as determined by the location of each sample) and plotted them in a biplot with the seven most influential species (average contribution >2.5%) superimposed. For benthic community composition, we followed a similar process. After our initial categorization, we first combined live coral categories into "branching" and "other" and omitted all categories with fewer than three records across the entire dataset (bleached coral and hydroids) from the data. We also excluded the "unidentifiable" category (<5% of points). We then calculated the proportional contribution of each category to the benthos in a given sampled outcrop and arranged the data into a sample-by-category matrix and performed another nMDS analysis as per above (with square-root transformed data). We also performed a PERMANOVA and visualized the data in the same way as described above, but we did not perform the SIMPER routine due to the lower number of categories. Further, we scaled the size of the symbols to represent the percent of live coral cover. Finally, we statistically compared live coral cover among the two locations using a Bayesian hierarchical model. We logit-transformed proportional *LiveCoralCover* and specified *Location* as a fixed effect, with *Site* specified as a random effect. Model and chain specifications were programmed as described above.

To compare intrinsic temperature tolerances, as derived from $CT_{min}$ and $CT_{max}$ trials, we ran two separate Bayesian linear models. For both models, we specified an effect of *Population* (i.e., separate levels for each species and their respective Arabian Gulf and Gulf of Oman populations) on the critical thermal limit of individuals and examined differences between pairwise levels using post-hoc contrasts (Supplementary Tables 3 and 4). We also explored effects of body size on thermal tolerance but found no meaningful effect. Models were run with a Gaussian error distribution and the same specifications as the previous models (e.g., burn-in, iterations, priors, etc.). We took 1000 draws from the posterior parameters to fit posterior distributions as well as their mean and confidence bands and plotted them alongside the raw data. Furthermore, to examine location-specific differences in length−weight relationships and species-specific abundances, we isolated individuals from three species (*C. anomolus, E. pulcher*, and *E. ventermaculus*) and ran separate models for each species to test the effects of total length (TL) and *Location* on *Weight*, with log-transformations of both *Weight* and *TL* and the effect of location (with a random effect of *Site*) on abundance. We used a Gaussian error distribution for the first set of models since the data were continuous and approximately normally distributed. We used a negative binomial error distribution for the second set of models since the data were non-negative integers and over-dispersed when run under a Poisson distribution. To validate the model performance, we used the posterior parameters to predict values across a sequence of 100 evenly spaced values within the sampled size range of the two populations. We performed this 500 times and plotted each predicted model fit alongside the raw data. Models were run with the same prior and chain specifications as detailed above.

We examined prey item ingestion of the examined fishes using a network theory approach for both the COI and 23S markers[124]. We first created a presence−absence matrix of OTUs/ESVs across fish individuals in all species and their populations, creating a bipartite dietary network based on prey presence or absence. To examine the community structure within the network, we omitted all prey items with only a single occurrence across the dataset since the full dataset identified the majority of individuals as unique modules. This step reduced the COI dataset from 1357 to 1046 unique predator−prey interactions and the 23S dataset from 7872 to 5698 predator−prey interactions. We then sought to identify modules within the network using Newman's modularity measure[125]. We used Beckett's community detection algorithm[126], which we re-iterated 20 times for each dataset. We then used the convergent output from the 20 iterations to determine the module membership of each individual in our network. We then created a data frame from the original presence−absence matrix that contained each OTU/ESV and its linkage to the fish individual in two columns, which we then summarized by the respective modules. This created a list of symbolic edges in the network across the two columns, linking each prey item to a module, which we plotted as a bipartite dietary network tree using the Fruchterman−Reingold algorithm. We also plotted module membership in a mosaic plot.

Furthermore, for the COI and 23S markers, we investigated prey item diversity ingested by each species' population by producing interpolated and extrapolated rarefaction curves, which showcase sequencing depth by plotting prey item species richness by the total number of sequences detected for each species. We ran rarefaction analyses by rarefying species richness estimates for each species or population to an endpoint defined by the maximum sequences in any population using 100 bootstraps and 50 knots along the x-axis[127].

Finally, we modeled growth and mortality dynamics in cryptobenthic fish assemblages from the two locations, ultimately yielding three rate-based metrics that serve as indicators of energy and nutrient fluxes, thus indicating ecosystem functioning[21]: produced biomass (in g d$^{-1}$ m$^{-2}$), consumed biomass (in g d$^{-1}$ m$^{-2}$), and total turnover (% d$^{-1}$)[98,128,129]. Produced biomass represents the amount of fish tissue accumulated by an assemblage (in this case, a cryptobenthic fish assemblage collected in a given sample), thus considering only the growth that will occur on any given day (based on yearly averages in this case). Consumed biomass represents the amount of fish tissue that perished based on our estimates of fish mortality. In this pathway, the energy and nutrients produced by fishes are provided to other consumers or decomposers via predation or detritivory. Finally, total turnover expands on the classic estimate of turnover (the production/standing biomass [P/B] ratio[130]) by also including consumed biomass (consumed biomass/standing biomass)[128]. As such, the turnover metric approximates the rate at which particles flow through the system, either via incorporation into fish biomass or release to other consumers through mortality.

For the modeling, we first accrued species-specific information on maximum lengths and a range of coarse ecological traits (pertaining to diet, sociality, habitat association, and prevailing mean sea surface temperatures [SST]) from the literature for each species in our samples. We also extracted length−weight relationships at the family-level, since not all species in our samples were common enough to construct robust length−weight relationships. We then used these data to calculate species-specific growth coefficients ($K_{max}$) to the specified maximum size and modeled individual weight gain based on changes in fish size per day under a Von Bertalanffy Growth Model (VBGM)[129]. By subtracting the observed fish size (as obtained from our samples) from the weight obtained by the same fish after 1 day (from the model), we calculated the expected biomass production by that individual. We estimated daily mortality rates by calculating species-level mortality risk coefficients via VBGM parameters and SST[128,131], and then we adjusted the risk based on relationships between mortality and body size[132]. Using these coefficients, we obtained a daily survival probability for a given individual in the dataset. By combining this probability with biomass production as obtained from the previous step, we were able to generate the expected loss of biomass due to natural mortality at the individual level. Finally, we summed the individual-level estimates of weight, growth, and mortality for each sample to obtain community-level values of standing biomass, produced biomass, and consumed biomass, which we used to calculate total turnover as the combined quotients of produced and consumed biomass and standing biomass[133].

All data preparation, analyses, and visualizations were performed in R[134] (version 3.6.1) using the *tidyverse*[135], *vegan*[136], *brms*[123], *iNEXT*[127], *igraph*[137], *bipartite*[138], *tidybayes*[139], *xgboost*[140], *emmeans*[141], *oceanmap*[142], *ncdf4*[143] and *raster*[144] packages. All graphs were made using the *Trimma lantana* and *Coryphaena hippurus* color palettes and silhouettes in the package *fishualize*[145]. Growth modeling was performed using an alpha version of the package *rfishprod*[133].

**Reporting summary.** Further information on research design is available in the Nature Research Reporting Summary linked to this article.

## Data availability

All raw data necessary to reproduce the results are available along with raw photographs and temperature data on figshare (https://figshare.com/projects/Cryptobenthic_fish_assemblages_in_the_United_Arab_Emirates/81644).

## Code availability

All code necessary to reproduce the analyses is available on figshare (https://figshare.com/projects/Cryptobenthic_fish_assemblages_in_the_United_Arab_Emirates/81644) and the lead author's GitHub (https://github.com/simonjbrandl/UAE18-crypto-communities).

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

## Acknowledgements

We thank the Environment Agency Abu Dhabi (TMBS/18/L/179) and Dibba Municipality (unnumbered) for collection permits and the UAE Ministry of Environment and Climate Change for the tissue export permit (AUD-Q-22-1110520). All work was performed under NYUAD IACUC approval 18-0003. We further thank the NYU Abu Dhabi Center for Genomics and Systems Biology for sequencing funding and the NYU Abu Dhabi Core Facilities group for support of field collections and thermal experiments. We thank D. McParland and G. Vaughan for field support, N. Al-Mansoori for assistance with processing specimens in the laboratory, and K. Maslenikov and J. Huie for assistance in cataloging specimens at the University of Washington. Partial fieldwork funding was provided to L. Tornabene by the University of Washington. We thank B.S. Cheng and L.M. Komoroske for comments on our manuscript.

## Author contributions

S.J.B. and J.L.J. designed the study; S.J.B., J.L.J., J.M.C., and L.T. performed field collections; J.L.J. ran physiological trials; S.J.B., J.M.C., and L.T. performed laboratory work; J.A.B. and L.T. provided funding and resources; S.J.B. performed data analysis and visualization; S.J.B. and R.A.M. performed population modeling; S.J.B. wrote the first draft of the manuscript, and all authors contributed to writing thereafter.

## Competing interests

The authors declare no competing interests.
