## [Peer Review File · Nature Communications]

Reviewers' Comments:

Reviewer #1:

Remarks to the Author:

This study makes comparisons of fish communities between two gulfs and finds them to be different. It then inappropriately attributes these differences to differences in temperature regimes between the two locations. Two communities can differ for any number of reasons, temperature is just one. To conclude, on the basis of comparing just two sites, that temperature is the principal driver (from the Abstract: "Rather, impoverished body conditions of 35 populations in the Arabian Gulf point toward an increased energetic costs of growth and 36 homeostasis at higher temperatures). The scale at which the factor of interest varies here is "Gulf", consequently the unit of replication to make valid inferences is also "Gulf". The study does not replicate at that scale and so none of the inferences are supported. I recognise that replicating at that scale would be challenging but consider that that the statement that the authors is making is substantial - that temperature is the most important driver of differences among gulf - to make a substantive statement requires substantive evidence.

There is vast literature on making comparisons across large spatial scales such as the one proposed here, particularly protected area studies on both land and sea. Likewise, there is a large body of work on local adaptation of physiology (Conover in fish is a good exemplar). This work does not approach this problem at the appropriate scale and does not provide sufficient support for any of the key inferences.

I'm sorry I cannot be more positive but for work to be published in a high impact journal, best practice in terms of experimental design is essential.

Reviewer #2:

Remarks to the Author:

This study compares cryptobenthic reef fish communities and their ecological processes from the Gulf of Oman and the Arabian Gulf. The Arabian Gulf is an extreme environment and this study investigating how this influences these fish in the context of future climate change. Through standardized collections and aquaria trials, the authors explored differences in, critical thermal maximum and minimum tolerances, gut content using DNA barcoding, and length weight relationships for a subset of species. They also modeled growth, mortality, and biomass turnover for each region at the community level. Through these

analysis, the authors conclude that there are ecological differences between the fish in the Arabian Gulf and Gulf of Oman reefs driven by extreme temperature after coral cover and composition are dismissed, with the extreme temperatures in the Arabian Gulf having a detrimental effect on fish communities and function.

Firstly, I would like to acknowledge the significant amount of field, laboratory, and aquaria work to address these questions. There is a considerable amount of effort gone into executing this study in a region that is greatly understudied and challenging to work in. This study is excellently communicated, analysed, and presented. Furthermore, the majority of this data is collected from the natural environment in natural conditions which is rare. However, this does reduce the ability to account and control for other variables. They have done a great job comparing the two regions, however I caution the strong narrative that focuses on temperature as the main driver.

The authors propose that the higher temperatures in the Arabian Gulf is the main driving difference between the two regions after the benthos is excluded. This is a focus of the manuscript and made clear in the title and throughout the manuscript (L130-140, Fig 1). While temperature profiles are clearly different and a driver of organism response, there appears to be no inclusion and little discussion around other environmental variables that differ greatly between regions that could factor into these findings (e.g., salinity and productivity).

The authors do test and dismiss benthic community composition and coral cover after finding no significant differences between regions based on the sample sites. Given the broad resolution of the benthic composition data, these differences might be difficult to detect. Resolution (benthic species, caves, complexity) could be important given the close relationship between fish and benthos for shelter, feeding and reproduction. Benthic profiling (5 x 20x20cm quadrates) over a ~4.6m² area with broad groups might not capture enough of the variation to detect differences in benthic composition and coral cover in addition to the assumption of the reef surface area (L657), and therefore dismiss the benthos as an additional driver. Given the extreme environment in the Arabian Gulf and the bleaching events in the previous years (L448) it is surprising that they are similar (see Burt et al. 2019 CR).

Sampling and collections were conducted at the end of April and May, the end of the cooler season. If the model species in this study only live weeks to months (Lines 99 onwards: 7.4 generations per year, high mortality) and you collected

them at the end of the winter, would it not be the winter temperatures that are influencing body mass per unit length, growth rates, and biomass production?

Minor comments

L130

Here you are presenting a maximum and minimum sea surface temperature from a 10 year period (2008 - 2018). While this presents the extremes for each region from a decade, some additional information about the two regions would be useful. Either here, or in the methods (L475), can you provide some additional details around the thermal regimes: i.e., how long do these max. and min. temps persist for, are these temps observed most years, and what is the average temperature for each of the two regions?

L154

It would be informative to include some values for species richness and density within this section. Also, the graph has "richness", the caption has "density", and the text has "diversity".

L198

Please provide the values for each region or the difference in heat tolerance for *E. pulcher*

L327

This is a strong statement considering only one species increased max thermal tolerance and one species increased min thermal tolerance.

L324

While statistically significant, how ecologically relevant are the differences in body condition for *E. ventermaculus* (very small weights) and *C. anomolus* (for the given size comparisons)?

L330 – 333

Poor body condition suggests a response to temperature and other environmental variables, - what about the recorded differences in diet composition (as a result of the temperature and other environmental variables) also producing poor body condition?

L338 – 342 and L423 - 426

How does the difference in produced biomass between regions related to the

observed standing biomass that was similar between the two regions?

L350, L352

If there is significant self-recruitment back to these thermal hotspots.

L382

Maybe add that DNA metabarcoding cannot accurately assess quantity of prey types in the diet.

L408

Your length body rate relationships are potential from winter months (if their life span is ~3 months). Therefore the response could be to cooler water not warmer water, or the temperature envelope if they live longer than a year. Would you expect these differences to be even more pronounced if they were collected at the end of the summer period? What is the average life span of these 3 species that you tested?

L485

What was the local water temperatures for each region at the time of the collections?

L517-520

Please provide sample size for each species

L554 - 557

Please include what the temperature and salinity for the holding tanks and trial chambers, and was this the same for species from each region?

L565

Can you please provide the specific replication for each species from each region for each thermal trial (max and min).

L657

"assumption that outcrops are hemispherical... " This seem like a big assumption, and assumes that all outcrops were of similar shape (and height?). Given the collection tarpaulin was standardized and used in this equation, it would appear that surface area would be similar among sites. This can obviously influence benthic groups % cover, in addition to calculating abundance, diversity, and biomass of fish based on the area of this calculation.

L671

Please include if the data was transformed for any of the ordination analysis?

L680

Please include if this is proportion or percentage.

L682

Please define if these "3 records" were per quadrat, outcrop, or region?

L694

Did you account for the effect of individual size?

Fig 1 and 2

It is difficult to distinguish among symbols.

Fig 2

Should the % coral cover in the bottom right be proportion? 0.2% to 0.6% seem extremely low

Fig 3

Check that 11.9°C is correct in the caption.

Reviewer #3:

Remarks to the Author:

The manuscript titled 'Organismal responses to extreme temperature reduce coral reef biodiversity and functioning' is study on differences in cryptobenthic fish communities among two reefs with different environmental conditions. The aim here is to understand how and why a community might look like in the light of increasing temperature due to climate change.

General comments:

What a lovely manuscript to review. The authors have done a great job at finding a good 'natural laboratory' to answer some of these really important questions. The study has been thoroughly performed and addresses many aspects of why fish might be different among both sites.

I only have one more general comment, which I can see in the discussion that the authors are already aware of. Although benthic communities might be similar, much other than temperature will vary among the 'two' locations. Especially the introduction is heavily focused on temperature. I think it is worth

already at the start to acknowledge that there is more than just temperature that differs between these sites. It is obviously impossible to find two sites with only one variable of change of course, but here temperature is the only variable presented.

Specific comments:

As mentioned to the editor, some of your study does not fall into my field of expertise, nevertheless it would be good to clarify some of these questions I stumbled across.

Line 187-190: I had to read this over and over again. It's a nicely compact sentence, but hard to follow, especially at this point it is the first time the reader hears of CTmax

In the next paragraphs it is not 100% clear if these species are found in both sites or just at one or the other (unless stated in brackets).

The issue with having a methods part after the results – according to your figure 3 – you measured two species that occur in both locations, one that is AG only and three GoO only? Maybe clarify this in the text (as it took me a while to figure out) and the 'all species' in line 191 might be misleading – it is all species which you tested (correct?).

Line 284: Is the fact that you only found large individuals of *C. anomalus* in the AG a collection bias or are there no large sized fish?

Line 335: Is there a chance that the fish at AG feed on restricted food – not just because there is less available - but possibly the food they choose to eat is of higher 'quality'(and hence less diverse)? (ok, saw that you did discuss this a few paragraphs later)

Line 329: very cool result.

Line 352: what about rapid evolution? Kind of curious to see if these are still the same species or there is some population structuring and local adaptation happening (but of course out of the scope of this paper).

Figure 1: Would it be possible to zoom into sites (as in the end the whole AG temp profile isn't really that important here)?

Figure 2: It is hard to identify the sites as the symbols are too small.

REVIEWER COMMENTS

Reviewer #1 (Remarks to the Author):

This study makes comparisons of fish communities between two gulfs and finds them to be different. It then inappropriately attributes these differences to differences in temperature regimes between the two locations. Two communities can differ for any number of reasons, temperature is just one. To conclude, on the basis of comparing just two sites, that temperature is the principal driver (from the Abstract: "Rather, impoverished body conditions of 35 populations in the Arabian Gulf point toward an increased energetic costs of growth and 36 homeostasis at higher temperatures). The scale at which the factor of interest varies here is "Gulf", consequently the unit of replication to make valid inferences is also "Gulf". The study does not replicate at that scale and so none of the inferences are supported. I recognise that replicating at that scale would be challenging but consider that that the statement that the authors is making is substantial - that temperature is the most important driver of differences among gulf - to make a substantive statement requires substantive evidence. There is vast literature on making comparisons across large spatial scales such as the one proposed here, particularly protected area studies on both land and sea. Likewise, there is a large body of work on local adaptation of physiology (Conover in fish is a good exemplar). This work does not approach this problem at the appropriate scale and does not provide sufficient support for any of the key inferences. I'm sorry I cannot be more positive but for work to be published in a high impact journal, best practice in terms of experimental design is essential.

Our response: We thank Reviewer 1 for their assessment of our paper. In accordance with the other two reviewers, we have de-emphasized the direct effect of temperature in our results. This also addresses the reviewer's concerns that our sampling design is not appropriate to establish a general effect of temperature. Nevertheless, our replication within each location and the multiple lines of evidence presented clearly allow for the conclusion that the environmental extremes that characterize reefs in the Arabian Gulf are the main driver of the divergent patterns in fish assemblages. As such, we believe that our overall conclusions are well-supported and robust.

Reviewer #2 (Remarks to the Author):

This study compares cryptobenthic reef fish communities and their ecological processes from the Gulf of Oman and the Arabian Gulf. The Arabian Gulf is an extreme environment and this study investigating how this influences these fish in the context of future climate change. Through standardized collections and aquaria trials, the authors explored differences in, critical thermal maximum and minimum tolerances, gut content using DNA barcoding, and length weight relationships for a subset of species. They also modeled growth, mortality, and biomass turnover for each region at the community level. Through these analysis, the authors conclude that there are ecological differences between the fish in the Arabian Gulf and Gulf of Oman reefs driven by extreme temperature after coral cover and composition are dismissed, with the extreme temperatures in the Arabian Gulf having a detrimental effect on fish communities and function.

Firstly, I would like to acknowledge the significant amount of field, laboratory, and aquaria work to address these questions. There is a considerable amount of effort gone into to executing this study in a region that is greatly understudied and challenging to work in. This study is excellently communicated, analysed, and presented. Furthermore, the majority of this data is collected from the natural environment in natural conditions which is rare, However, this does reduce the ability to account and control for other variables. They have done a great job comparing the two regions, however I caution the strong narrative that focuses on temperature as the main driver.

Our response: Thank you for your constructive and insightful feedback. We are delighted that you found our study to be well executed. We understand the concerns regarding the strong inference of causal temperature effects in the absence of other environmental factors and have, therefore, modified our interpretation of the main drivers accordingly.

The authors propose that the higher temperatures in the Arabian Gulf is the main driving difference between the two regions after the benthos is excluded. This is a focus of the manuscript and made clear in the title and throughout the manuscript (L130-140, Fig 1). While temperature profiles are clearly different and a driver of organism response, there appears to be no inclusion and little discussion around other environmental variables that differ greatly between regions that could factor into these findings (e.g., salinity and productivity).

Our response: We appreciate this perspective on our results and agree. We have altered several parts of the manuscript accordingly. We have modified the title to prevent direct causal inference of temperature effects ("Extreme environmental conditions erode coral reef biodiversity and functioning") and have broadened our descriptions of the

environmental conditions, with repeated mention of other co-varying factors such as salinity, productivity, or dissolved oxygen. Nevertheless, we caution that temperature is widely seen as the dominant environmental force that shapes life in the Gulf (see Sheppard et al. 1992; Riegl & Purkis 2012; Hume et al. 2013; Howells et al. 2016) and, as such, we do maintain that temperature is likely to be the dominant driver of the observed patterns. We hope that the revised version provides a more balanced account.

The authors do test and dismiss benthic community composition and coral cover after finding no significant differences between regions based on the sample sites. Given the broad resolution of the benthic composition data, these differences might be difficult to detect. Resolution (benthic species, caves, complexity) could be important given the close relationship between fish and benthos for shelter, feeding and reproduction. Benthic profiling (5 x 20x20cm quadrates) over a ~4.6m² area with broad groups might not capture enough of the variation to detect differences in benthic composition and coral cover in addition to the assumption of the reef surface area (L657), and therefore dismiss the benthos as an additional driver. Given the extreme environment in the Arabian Gulf and the bleaching events in the previous years (L448) it is surprising that they are similar (see Burt et al. 2019 CR).

Our response: We agree with the reviewer that this finding is surprising. Based on our in-situ collections, we expected benthic community composition to be more distinct, but the data showed substantial overlap, at least in the coarse categories that we were able to annotate from our photographs. We now make it explicit that, despite this overlap, benthic configurations were distinct and may have a role in the observed patterns. However, given that the surface areas of the outcrops are approximately equal (despite assumptions of idealized hemispheres) and the relatively robust comparison of live coral cover, we believe that the exclusion of benthic configuration and live coral cover as major drivers of the observed differences in cryptobenthic fish assemblages is merited.

Sampling and collections were conducted at the end of April and May, the end of the cooler season. If the model species in this study only live weeks to months (Lines 99 onwards: 7.4 generations per year, high mortality) and you collected them at the end of the winter, would it not be the winter temperatures that are influencing body mass per unit length, growth rates, and biomass production?

Our response: This is a good point and requires some discussion. While our sampling did indeed occur during a more moderate time of year, our argument is based on observed changes in energy budgets through transgenerational adaptation to warmer temperatures. Donelson et al. (2010) and Bernal et al. (2018) showed that fishes reared throughout their life-cycles at elevated temperature had altered metabolic rates and body conditions, even at normal temperatures, than fishes reared at ambient temperatures throughout. Similarly, there is evidence for relatively rapid transgenerational plasticity in growth potential and body condition, where the parental environment influences offspring growth at a range of different environmental conditions (Salinas & Munch 2012). As such, we hypothesize that adaptations that permit the few species that occupy reefs in the Arabian Gulf to persist, are only able to do so at the expense of their overall condition.

Minor comments

L130

Here you are presenting a maximum and minimum sea surface temperature from a 10 year period (2008 - 2018). While this presents the extremes for each region from a decade, some additional information about the two regions would be useful. Either here, or in the methods (L475), can you provide some additional details around the thermal regimes: i.e., how long do these max. and min. temps persist for, are these temps observed most years, and what is the average temperature for each of the two regions?

Our response: As requested, we now provide more information on the remotely sensed data as well as in situ data from temperature loggers. This information is now included in the text and supplemental material (Figs. S1,2; Table S1).

L154

It would be informative to include some values for species richness and density within this section. Also, the graph has "richness", the caption has "density", and the text has "diversity".

Our response: We have provided the requested values and have made the term "richness" consistent throughout. Thank you for pointing this out.

L198

Please provide the values for each region or the difference in heat tolerance for *E. pulcher*

Our response: Done as suggested.

L327

This is a strong statement considering only one species increased max thermal tolerance and one species increased min thermal tolerance.

Our response: We agree and have now toned down this statement.

L324

While statistically significant, how ecologically relevant are the differences in body condition for *E. ventermaculus* (very small weights) and *C. anomolus* (for the given size comparisons)?

Our response: We now provide the difference in weight at the mean length of the three species as predicted from the model fits. These values show that, at their mean total length, E. ventermaculus, E. pulcher, and C. anomolus are 67.2%, 62.2% and 10.0% heavier in the Gulf of Oman compared to the Arabian Gulf. As such, we believe that this suggests a significant ecological effect, at least for E. ventermaculus and E. pulcher. We interpret the effect for C. anomolus more cautiously, which is in line with our previous interpretation.

L330 – 333

Poor body condition suggests a response to temperature and other environmental variables, - what about the recorded differences in diet composition (as a result of the temperature and other environmental variables) also producing poor body condition?

Our response: This is an excellent point. We have amended our interpretation to be more inclusive of the dietary differences (e.g. l. 566; l. 601).

L338 – 342 and L423 - 426

How does the difference in produced biomass between regions related to the observed standing biomass that was similar between the two regions?

Our response: The lack of differences in standing biomass is largely due to the stochastic presence of single individuals of larger fishes in the Arabian Gulf (e.g. Pomacanthus maculosus, Colletteichthys sp.). While such individuals significantly contribute to standing biomass, they have a smaller part in the production and recycling of consumable biomass (e.g. Brandl et al. 2019).

L350, L352

If there is significant self-recruitment back to these thermal hotspots.

Our response: Good point. Added as suggested.

L382

Maybe add that DNA metabarcoding cannot accurately assess quantity of prey types in the diet.

Our response: Done as suggested.

L408

Your length body rate relationships are potential from winter months (if their life span is ~3 months). Therefore the response could be to cooler water not warmer water, or the temperature envelope if they live longer than a year. Would you expect these differences to be even more pronounced if they were collected at the end of the summer period? What is the average life span of these 3 species that you tested?

Our response: Thank you, this is an excellent point that requires clarification. As mentioned above, we now specify that the temperature-based response would act on the sampled individuals via transgenerational changes that are induced by cross-generational exposure to the extreme conditions in the Arabian Gulf. Most cryptobenthics will indeed not live longer than one year, but there is no direct age estimate for the sampled species. On average, blennies (such as Ecsenius pulcher) have an average maximum lifespan of 1.79 years, gobies (such as Coryogalops anomolus) of 0.97, and triplefins (such as Enneapterygius ventermaculus) of 0.64 years (Brandl et al. 2018). Based on these estimates, we would assume some individuals live through multiple or even repeated seasons, but the vast majority will likely be less than one year old. It is certainly possible that sampling at the end of summer would further increase the difference through acute environmental stress (e.g. Nowicki et al. 2012), but at this stage, we do not have the data to investigate this question. It is worth highlighting, however, that the recorded differences in this study

are found at the most benign time of year for the Arabian Gulf (i.e. just before the most extreme season), suggesting that our results are conservative and differences are likely to be even more pronounced at the end of the summer.

L485

What was the local water temperatures for each region at the time of the collections?

Our responses: Temperatures at the time of sampling were between 27°C and 29°C in the Arabian Gulf and Gulf of Oman, respectively. We now provide these values in the methods.

L517-520

Please provide sample size for each species

Our response: We now provide sample sizes in Table S4.

L554 - 557

Please include what the temperature and salinity for the holding tanks and trial chambers, and was this the same for species from each region?

Our response: We now provide the parameters for the trials. All fishes were kept and tested at the same temperature (27°C – 28°C) and salinity regime (42 ppt).

L565

Can you please provide the specific replication for each species from each region for each thermal trial (max and min).

Our response: Yes, this is now provided in Table S4.

L657

“assumption that outcrops are hemispherical... “ This seem like a big assumption, and assumes that all outcrops were of similar shape (and height?). Given the collection tarpaulin was standardized and used in this equation, it would appear that surface area would be similar among sites. This can obviously influence benthic groups % cover, in addition to calculating abundance, diversity, and biomass of fish based on the area of this calculation.

Our response: You are correct that the idealized hemisphere is an assumption that is unlikely to be accurate across all samples. Nevertheless, as mentioned above, the dimensions of the net and tarpaulin constrain the size variations of each outcrop, resulting in broadly comparable surface areas for each sample. We hope to validate this in a future study that uses 3D-photogrammetry to estimate surface areas.

L671

Please include if the data was transformed for any of the ordination analysis?

Our response: Both datasets were square-root transformed before the ordinations. We now include this in the manuscript.

L680

Please include if this is proportion or percentage.

Our response: Done.

L682

Please define if these “3 records” were per quadrat, outcrop, or region?

Our response: This was across the entire dataset. We now state this in the manuscript.

L694

Did you account for the effect of individual size?

Our response: We initially included body mass as a variable in the model, but we removed it after revealing no effect. We now specify this in the manuscript.

Fig 1 and 2

It is difficult to distinguish among symbols.

Our response: We have increased the size of the symbols.

Fig 2

Should the % coral cover in the bottom right be proportion? 0.2% to 0.6% seem extremely low

Our response: Well caught. This should be 20%, 40%, and 60%. Thank you.

Fig 3

Check that 11.9oC is correct in the caption.

Our response: Checked. This should read "Average critical thermal minimum" – well spotted.

Reviewer #3 (Remarks to the Author):

The manuscript titled 'Organismal responses to extreme temperature reduce coral reef biodiversity and functioning' is study on differences in cryptobenthic fish communities among two reefs with different environmental conditions. The aim here is to understand how and why a community might look like in the light of increasing temperature due to climate change.

General comments:

What a lovely manuscript to review. The authors have done a great job at finding a good 'natural laboratory' to answer some of these really important questions. The study has been thoroughly performed and addresses many aspects of why fish might be different among both sites.

Our response: Thank you for your kind and constructive comments on our manuscript.

I only have one more general comment, which I can see in the discussion that the authors are already aware of. Although benthic communities might be similar, much other than temperature will vary among the 'two' locations. Especially the introduction is heavily focused on temperature. I think it is worth already at the start to acknowledge that there is more than just temperature that differs between these sites. It is obviously impossible to find two sites with only one variable of change of course, but here temperature is the only variable presented.

Our response: This is a fair point, and we have adjusted our language to be more cautious about the causal effects of temperature throughout the manuscript.

Specific comments:

As mentioned to the editor, some of your study does not fall into my field of expertise, nevertheless it would be good to clarify some of these questions I stumbled across.

Line 187-190: I had to read this over and over again. It's a nicely compact sentence, but hard to follow, especially at this point it is the first time the reader hears of CTmax

Our response: We have modified and clarified the sentence by omitting the first section.

In the next paragraphs it is not 100% clear if these species are found in both sites or just at one or the other (unless stated in brackets).

The issue with having a methods part after the results – according to your figure 3 – you measured two species that occur in both locations, one that is AG only and three GoO only? Maybe clarify this in the text (as it took me a while to figure out) and the 'all species' in line 191 might be misleading – it is all species which you tested (correct?).

Our response: We have now included more information; we directly state which species were tested for intraspecific plasticity and for which species we only sampled single populations.

Line 284: Is the fact that you only found large individuals of *C. anomalus* in the AG a collection bias or are there no large sized fish?

*Our response: As far as we are aware, there were no larger sized *C. anomalus* found in the Gulf of Oman. Given that we sampled the area extensively, we believe that this is a real pattern.*

Line 335: Is there a chance that the fish at AG feed on restricted food – not just because there is less available - but

possibly the food they choose to eat is of higher 'quality'(and hence less diverse)? (ok, saw that you did discuss this a few paragraphs later)

Our response: Yes, absolutely. We're happy to hear that you came to the same conclusion.

Line 329: very cool result.

Our response: Thank you!

Line 352: what about rapid evolution? Kind of curious to see if these are still the same species or there is some population structuring and local adaptation happening (but of course out of the scope of this paper).

Our response: We would be equally curious to investigate this potential.

Figure 1: Would it be possible to zoom into sites (as in the end the whole AG temp profile isn't really that important here)?

Our response: We chose to retain the full map since we think it is useful to show the rest of the Arabian Gulf. However, we now provide extended temperature data to provide more details.

Figure 2: It is hard to identify the sites as the symbols are too small.

Our response: We have increased the size of the symbols. Thank you again for your constructive comments.

Reviewers' Comments:

Reviewer #2:

None

Reviewer #3:

Remarks to the Author:

The manuscript titled 'Organismal responses to extreme temperature reduce coral reef biodiversity and functioning' has now been revised and the authors addressed my main concern about possible other factors but temperature playing a role here. It is a massive piece of work and adds an important puzzle piece to how coral reefs might 'adjust' to future environmental changes.

I only have small comments and one comment on a part of the discussion that was changed since the first review:

Line 54: Not sure 'extreme' is the correct word. You could refer to environmental 'changes' or environmental

Line 101: food availability?

Line 284: 'notable'

Line 396: Big jump here – provide an introductory sentence?

Line 400: at what depth was this temperature measured?

Line 424: the conditions were unfavorable also in the past or is there a change in temperature for that area more recently?

Line 512: you just mention five species by name above – why is it 6 in total? You could add anomolus to line 509. "And one species specific to the Arabian Gulf".

Line 690: 'coral' reef.

Line 832-392:

I am not sure I follow this argument:

Genetic adaptation and transgenerational plasticity isn't necessarily an either – or scenario. Ultimately adaptation needs to happen through allele frequency changes – but plasticity (transgenerational or not) can allow for survival especially in rapidly changing environmental conditions. I don't think that the fact that you found lesser body condition at the hotter reefs is evidence of transgenerational acclimation not playing a role here.

I had a quick look at the three papers that the authors mentioned in response to another reviewer (who asked about the fact that the body condition was based on winter temperatures) and I don't see that those studies look at the fish going back to a lower temperature (after high temperature exposure). In particular for a short-lived species experiencing very different environmental conditions through even a year from generation to the next my guess (!) would

be that carry-over effects, parental effects and transgenerational plasticity would play a role as well. That being said, it is obvious that lesser amounts of food (or worse quality) might be a more dominating issue.

It is logical that *C. anomalus* might be doing better as it has ancestors that live in more extreme conditions, hence, possibly having a genetic background that is favorable to extreme temperature conditions. However, the argument about resource limitation would also need to be considered here - does the genetic background aid the species to forage better - live better of less food availability/restricted food etc.?

Response to referees:

REVIEWERS' COMMENTS:

Reviewer #3 (Remarks to the Author):

The manuscript titled 'Organismal responses to extreme temperature reduce coral reef biodiversity and functioning' has now been revised and the authors addressed my main concern about possible other factors but temperature playing a role here. It is a massive piece of work and adds an important puzzle piece to how coral reefs might 'adjust' to future environmental changes.

Our response: Thank you again for the thoughtful feedback. We are very pleased with the reviewer's positive assessment of our work.

I only have small comments and one comment on a part of the discussion that was changed since the first review:

Line 54: Not sure 'extreme' is the correct word. You could refer to environmental 'changes' or environmental

Our response: changed to "environmental change"

Line 101: food availability?

Our response: changed as suggested.

Line 284: 'notable'

Our response: done.

Line 396: Big jump here – provide an introductory sentence?

Our response: done.

Line 400: at what depth was this temperature measured?

Our response: between 4-6m depth. We have added this information.

Line 424: the conditions were unfavorable also in the past or is there a change in temperature for that area more recently?

Our response: to our knowledge, conditions have been unfavorable throughout the history of the Arabian Gulf. We now mention this.

Line 512: you just mention five species by name above – why is it 6 in total? You could add anomolus to line 509. “And one species specific to the Arabian Gulf”.

Our response: done as suggested.

Line 690: ‘coral’ reef.

Our response: done.

Line 832-392:

I am not sure I follow this argument:

Genetic adaptation and transgenerational plasticity isn't necessarily an either – or scenario. Ultimately adaptation needs to happen through allele frequency changes – but plasticity (transgenerational or not) can allow for survival especially in rapidly changing environmental conditions. I don't think that the fact that you found lesser body condition at the hotter reefs is evidence of transgenerational acclimation not playing a role here. I had a quick look at the three papers that the authors mentioned in response to another reviewer (who asked about the fact that the body condition was based on winter temperatures) and I don't see that those studies look at the fish going back to a lower temperature (after high temperature exposure). In particular for a short-lived species experiencing very different environmental conditions through even a year from generation to the next my guess (!) would be that carry-over effects, parental effects and transgenerational plasticity would play a role as well. That being said, it is obvious that lesser amounts of food (or worse quality) might be a more dominating issue. It is logical that *C.anomalous* might be doing better as it has ancestors that live in more extreme conditions, hence, possibly having a genetic background that is favorable to extreme temperature conditions. However, the argument about resource limitation would also need to be considered here - does the genetic background aid the species to forage better – live better of less food availability/restricted food etc.?

Our response: this is a very good point. We have addressed this by revising our statements concerning adaptation/transgenerational plasticity, which we provide no evidence for. We now refer predominantly to “intraspecific shifts in thermal tolerances,” which adequately captures the results but is less speculative concerning the underlying microevolutionary mechanisms. We believe that this is more appropriate and thank the reviewer for this sage comment.